# The effects of chloride dynamics on substantia nigra pars reticulata responses to pallidal and striatal inputs

**Ryan S Phillips[1,2], Ian Rosner[2,3], Aryn H Gittis[2,3], Jonathan E Rubin[1,2]***

[1]Department of Mathematics, University of Pittsburgh, Pittsburgh, United States; [2]Center for the Neural Basis of Cognition, Pittsburgh, United States; [3]Department of Biological Sciences, Carnegie Mellon University, Pittsburgh, United States

**Abstract** As a rodent basal ganglia (BG) output nucleus, the substantia nigra pars reticulata (SNr) is well positioned to impact behavior. SNr neurons receive GABAergic inputs from the striatum (direct pathway) and globus pallidus (GPe, indirect pathway). Dominant theories of action selection rely on these pathways' inhibitory actions. Yet, experimental results on SNr responses to these inputs are limited and include excitatory effects. Our study combines experimental and computational work to characterize, explain, and make predictions about these pathways. We observe diverse SNr responses to stimulation of SNr-projecting striatal and GPe neurons, including biphasic and excitatory effects, which our modeling shows can be explained by intracellular chloride processing. Our work predicts that ongoing GPe activity could tune the SNr operating mode, including its responses in decision-making scenarios, and GPe output may modulate synchrony and low-frequency oscillations of SNr neurons, which we confirm using optogenetic stimulation of GPe terminals within the SNr.

## Introduction

The substantia nigra pars reticulata (SNr) is the primary output nucleus of the rodent basal ganglia (BG) and hence likely plays a key role in the behavioral functions, such as decision-making and action selection, suppression, or tuning, to which the BG contribute. The SNr exhibits intrinsic spiking activity, resulting in ongoing GABAergic outputs to specific thalamic sites, which are believed to suppress unwanted or spurious movements. While the literature on signal transmission through the basal ganglia emphasizes the projection from the subthalamic nucleus to the SNr, the SNr also receives converging GABA$_A$-receptor mediated synaptic inputs associated with the two major transmission channels through the BG, the direct and indirect pathways. Thus, the behavioral influence of the BG is ultimately regulated by how the SNr integrates these inputs.

Although dominant theories of action selection strongly rely on the inhibitory actions of these pathways on SNr, the details of this integration process have not been thoroughly investigated and remain poorly understood. Interestingly, the inputs to SNr from the two pathways feature distinct characteristics. Indirect pathway GABAergic projections to SNr arise from the external segment of the globus pallidus (GPe), which engages in tonic spiking activity; occur via basket-like synapses around the soma of SNr neurons; and exhibit short-term depression. Direct pathway inputs are delivered by striatal (Str) neurons, which spike much more sparsely; are located on distal dendrites; and exhibit short-term facilitation (*Smith and Bolam, 1991*; *von Krosigk et al., 1992*; *Connelly et al., 2010*; *Lavian and Korngreen, 2016*). The complexity of how these aspects interact may have hindered the study of the convergence of these inputs to the SNr, yet there may be an additional, easily overlooked factor influencing the process as well: GABA dynamics (*Raimondo et al., 2012*; *Doyon et al., 2011*). The ongoing activity of GPe neurons would likely induce a large tonic chloride

**\*For correspondence:**
jonrubin@pitt.edu

load on SNr neurons, potentially depolarizing the GABA reversal potential, $E_{GABA}$. Although striatal inputs are less frequent, their impacts would be affected by chloride accumulation, which could be exaggerated in smaller dendritic compartments, and by associated variability of $E_{GABA}$. Indeed, past studies have reported $E_{GABA}$ values that vary over a relatively wide range, from $-80$ to $-55$ mV, in SNr (*Giorgi et al., 2007*; *Connelly et al., 2010*; *Higgs and Wilson, 2016*; *Simmons et al., 2018*). Moreover, earlier experiments showed excitatory effects along with inhibitory ones from stimulation of SNr-projecting Str neurons in vivo (*Freeze et al., 2013*), which could relate to chloride regulation as well.

To study this complex combination of effects and their possible functional consequences, we developed a computational model of an SNr neuron including somatic and dendritic compartments and the corresponding GABAergic inputs as well as the dynamics of intracellular chloride and $E_{GABA}$. We used this model to investigate the influence of GABAergic synaptic transmission from GPe, Str, and SNr collaterals on SNr activity under behaviorally relevant conditions. We found that with the inclusion of short-term synaptic plasticity tuned to fit previous data, the model's dynamics matched a range of experimental findings on SNr firing patterns, including our own new results from optogenetic stimulation in mice. Given this agreement, we used the model to generate novel predictions about how direct and indirect pathway inputs may shape SNr activity patterns in functional settings involving both pathways. Specifically, we predict that variations in the level of GPe activity could interact with sparse SNr reciprocal interconnectivity to provide an effective mechanism to tune SNr synchrony and the emergence of low-frequency oscillations, and we present experimental data based on optogenetic stimulation of GABAergic GPe terminals in the SNr that provides evidence of this effect. We also predict that ongoing high-frequency GPe activity could serve a modulatory role in action selection by adjusting the effectiveness of lower-frequency direct pathway Str signals at pausing SNr outputs to downstream targets, as would be needed to allow action selection. The convergence of multiple GABA$_A$ receptor-mediated synaptic input streams onto individual neurons, such as pyramidal neurons in cortex, represents a common scenario in neural circuitry, and our results suggest that intracellular $Cl^-$ levels should also be considered in analyzing the integration of GABAergic inputs by neurons in brain regions beyond the SNr.

## Results

### Conductance-based SNr model

Due to the positioning of the SNr within the BG, synaptic integration of GABAergic projections from the direct (Str) and indirect (GPe) pathways in the SNr is likely a critical factor in BG function. Nonetheless, the effects of these two pathways on SNr activity are not well understood. Complicating matters, GPe and Str inputs form synapses on disparate locations on SNr neurons, undergo distinct short-term synaptic plasticity and likely have differing susceptibilities to breakdown of $E_{GABA}$, mediated by the $Cl^-$ load. Therefore, to investigate synaptic integration of GPe and Str GABAergic inputs to the SNr in more detail, we constructed a conductance-based neuron model with somatic and dendritic compartments (*Figure 1*). The two compartments are electrically coupled and intracellular $Cl^-$ concentration ($[Cl^-]_i$) is maintained in each compartment by the potassium-chloride co-transporter (KCC2). The baseline firing rate ($\approx$ 10 Hz) and action potential peak of the model are tuned to match experimental data from in vitro mouse and rat slice recordings (*Richards et al., 1997*; *Atherton and Bevan, 2005*; *Yanovsky et al., 2006*; *Zhou et al., 2008*; *Ding et al., 2011*), while the AHP is tuned to match data presented in *Higgs and Wilson, 2016*. For a full model description see *Materials and methods*.

### Short-term synaptic depression and facilitation of GPe and Str synaptic projections

The GABAergic synapses from the GPe and Str neurons undergo short-term synaptic depression and facilitation, respectively. To decide how to implement and tune these effects in our model, we turned to the experimental literature. Two studies reported on short-term plasticity of GPe and Str projections in in vitro slice preparations (*Connelly et al., 2010*; *Lavian and Korngreen, 2016*). Because this data was averaged over multiple neurons and trials, we incorporated an established mean-field model of short-term synaptic depression/facilitation (*Abbott, 1997*; *Dayan and Abbott,*

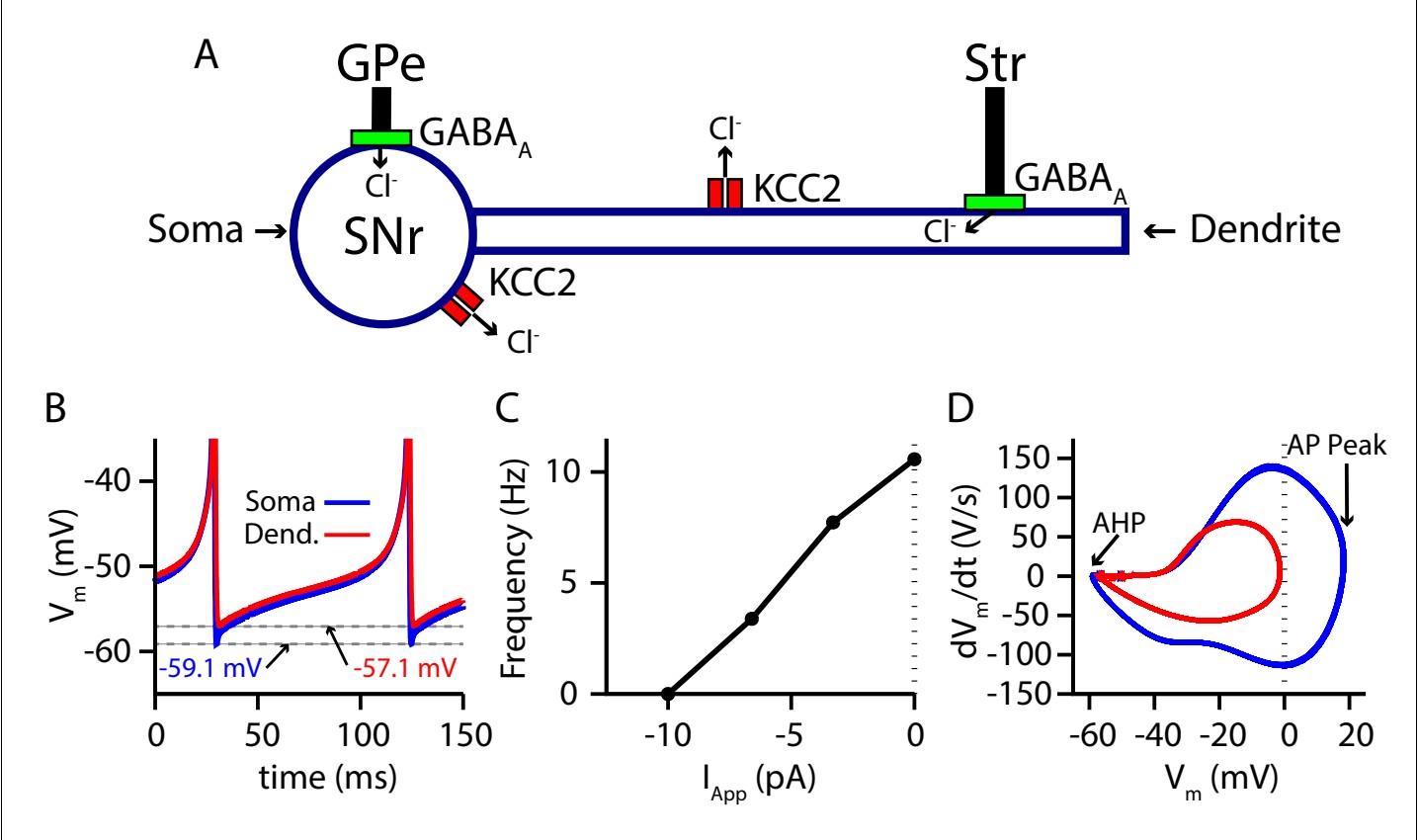

**Figure 1.** Two-compartment SNr model neuron includes currents that affect $[Cl^-]_i$ and produces appropriate dynamics. (A) Schematic diagram of the model. (B) Tonic spiking voltage traces for both compartments, with minimum voltages labeled. (C) Model f-I curve. (D) Phase plot of the rate of change of the membrane potential ($dV_m/dt$) against the membrane potential ($V_m$) showing afterhyperpolarization (AHP) and spike height (AP Peak) for both compartments.

*2001*; *Morrison et al., 2008*) into our simulated synaptic currents to capture short-term synaptic dynamics in our simulations.

Interestingly, the two experimental papers reported results that superficially appear to be at odds with each other. In *Connelly et al., 2010*, the magnitude of synaptic depression and facilitation of synapses onto SNr neurons was found to be largely independent of the tested stimulation frequencies (10 Hz, 50 Hz, 100 Hz). In contrast, in a BG output nucleus analogous to the SNr, the entopeduncular nucleus (EP), a similar characterization of the short-term synaptic dynamics of GPe and Str projections found that short-term depression and facilitation are highly frequency-dependent (*Lavian and Korngreen, 2016*). Moreover, the magnitude of synaptic facilitation of Str projections was shown to decrease in the EP for simulation frequencies above 10 Hz.

A critical distinction between these studies is that data was collected under a voltage-clamp configuration in *Connelly et al., 2010* and under a current-clamp configuration in *Lavian and Korngreen, 2016*. Under current-clamp, the membrane potential ($V_m$) is free to change. Consequently, stimulation of GPe or Str projections hyperpolarizes $V_m$ towards the GABAergic reversal potential ($E_{GABA}$), which reduces the GABAergic driving force ($V_m - E_{GABA}$) and ultimately decreases the magnitude of the inhibitory postsynaptic potential (IPSP). In contrast, the GABAergic driving force does not change under voltage-clamp, as $V_m$ is fixed. In both voltage- and current-clamp $E_{GABA}$ may also be considered fixed due to the whole cell configuration and free ionic diffusion between the cell and recording pipette. Based on these considerations, we tuned our model to match the voltage-clamp data from *Connelly et al., 2010*, as it is likely a better representation of the underlying short-term synaptic dynamics of GPe and Str inputs (*Figure 2*). Interestingly, with this tuning, the short-term GPe and Str synaptic dynamics in our model when tested under current-clamp also reproduces the synaptic dynamics reported in *Lavian and Korngreen, 2016*. Specifically, GPe synaptic depression

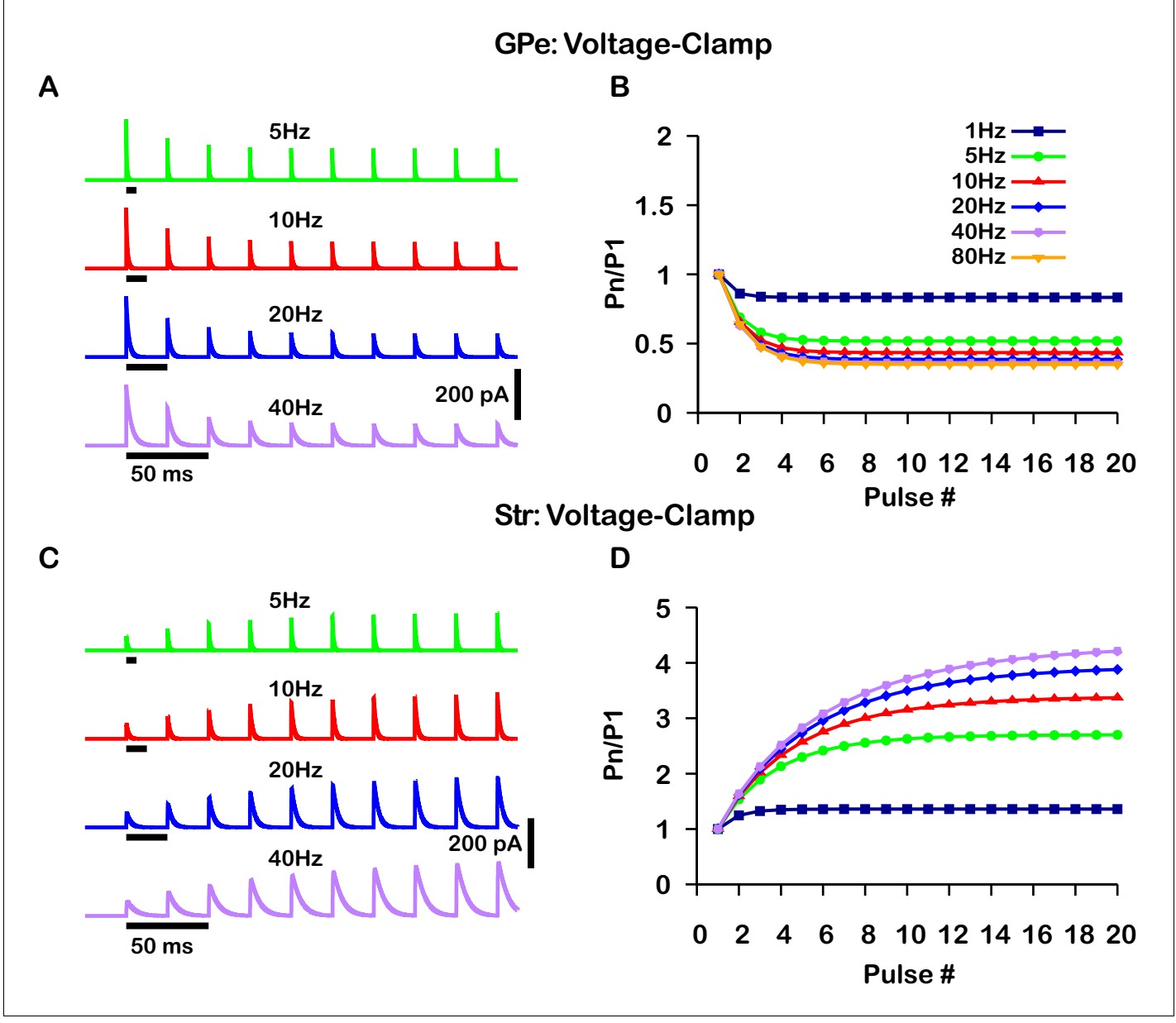

**Figure 2.** Simulated short-term synaptic depression and facilitation of GABAergic synapses originating from GPe neurons of the indirect pathway (A and B) and Str neurons of the direct pathway (C and D) under voltage clamp. For the GPe and Str simulations, the left traces (A and C) show current and right panels (B and D) show the pared pulse ratios (PPR) resulting from repeated synaptic stimulation at different frequencies. The amplitude of each IPSC ($P_n$) was normalized to the amplitude of the first evoked IPSC ($P_1$). For this set of simulations the membrane potential was held at $V_S = -60.0\,mV$ and $E_{GABA}$ for the somatic and dendritic compartments was held fixed at –72 mV. Model parameters and behavior were tuned to match voltage-clamp data from *Connelly et al., 2010*.

and Str synaptic facilitation are strongly frequency dependent, and the magnitude of synaptic facilitation in Str synapses decreases for stimulation frequencies above 10 Hz (*Figure 3*). These results demonstrate the importance of considering the differences between voltage- and current-clamp recordings when characterizing short-term synaptic dynamics. Additionally, these findings suggest that short-term synaptic dynamics of inputs from GPe and Str in the EP are tuned in a similar way to those in the SNr.

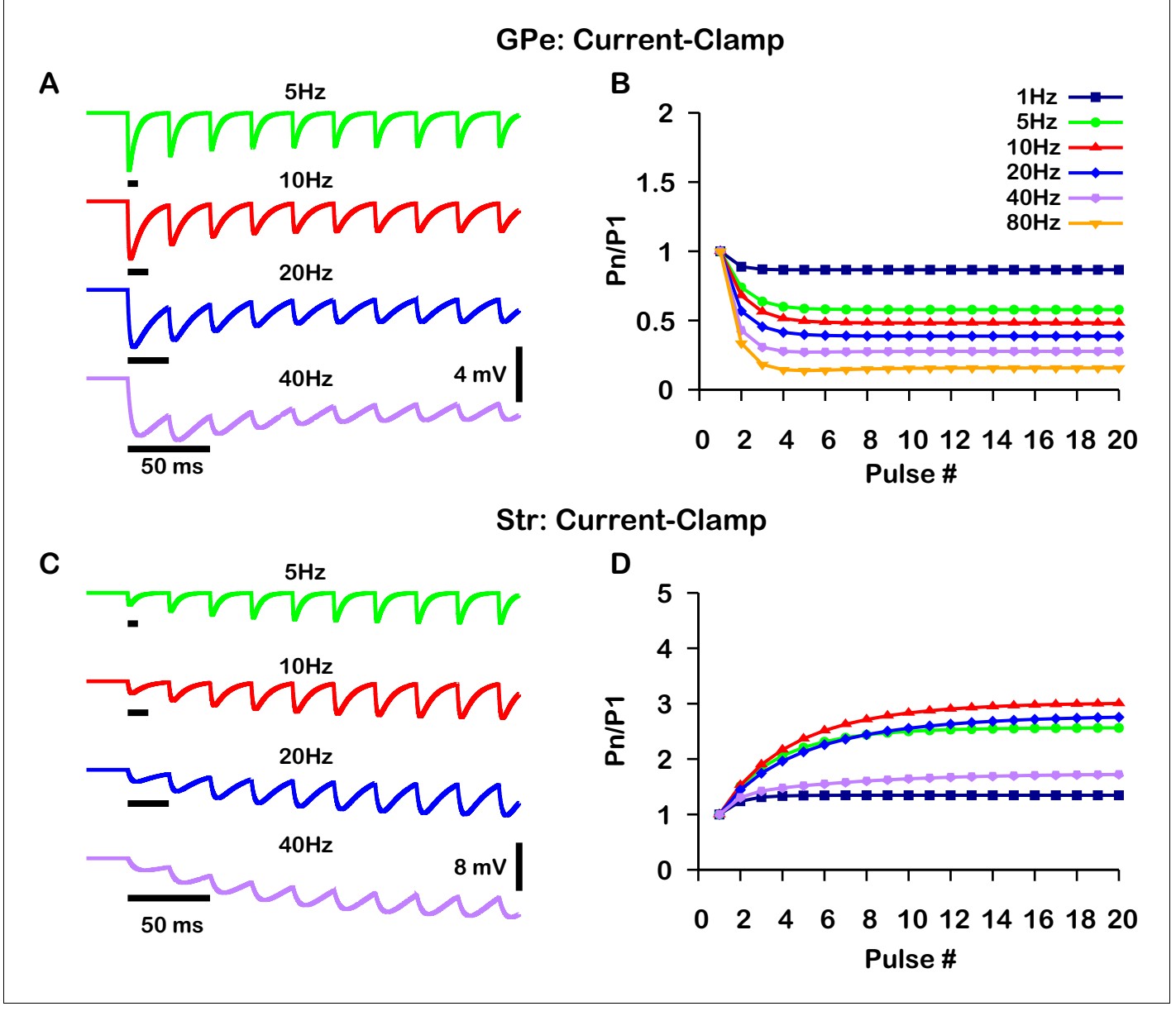

**Figure 3.** Simulated short-term synaptic depression and facilitation of GABAergic synapses originating from GPe neurons of the indirect pathway (A and B) and Str neurons of the direct pathway (C and D) under current clamp. For the GPe and Str simulations, the left traces (A and C) show voltage and right panels (B and D) show the pared pulse ratios (PPR) resulting from repeated synaptic stimulation at different frequencies. The amplitude of each IPSP ($P_n$) was normalized to the amplitude of the first evoked IPSP ($P_1$). For this set of simulations $I_{APP}$ was tuned to set the resting membrane of the somatic compartment at $V_S = -60.0\,mV$. In both compartments $E_{GABA}$ was held fixed at –70 mV. Model performance is qualitatively, and somewhat quantitatively, similar to experimental current-clamp data (*Lavian and Korngreen, 2016, Figures 2, 3*).

## SNr responses to simulated stimulation of GPe and Str inputs depend on $E_{GABA}$ and intracellular $Cl^-$ levels

Next, we used our model to consider effects of variability of the GABA reversal potential on SNr responses to its GABAergic inputs. Maintenance of the $Cl^-$ gradient is largely determined by a neuron's ability to preserve a low intracellular chloride concentration ($[Cl^-]_i$), which in turn depends on the balance of the neuron's capacity for $Cl^-$ extrusion by the potassium-chloride co-transporter KCC2 (*Doyon et al., 2011*; *Raimondo et al., 2012*; *Doyon et al., 2016*; *Mahadevan and Woodin,*

*2016*) and the $Cl^-$ influx into the neuron that occurs through $Cl^-$-permeable ion channels that contribute to $I_{GABA}$.

Due to the importance of $Cl^-$ regulation in GABAergic synaptic transmission, we first characterized the relationship among a conductance associated with a tonic chloride load ($g_{GABA}^{Tonic}$), the $Cl^-$ extrusion capacity ($g_{KCC2}$), and $E_{GABA}$ in the somatic compartment of our model (*Figure 4A*). We found that $E_{GABA}$ may vary from approximately –80 mV with very low net $Cl^-$ influx to approximately – 45 mV with high $g_{GABA}^{Tonic}$ and low $Cl^-$ extrusion capacity; note that the level of depolarization of $E_{GABA}$ is also influenced by the $HCO_3^-$ concentration gradient across the cell membrane (*Kaila and Voipio, 1987; Kaila et al., 1989; Staley et al., 1995; Staley and Proctor, 1999; Raimondo et al., 2012*; see *Materials and methods*, *Equation 23*). Importantly, depending on $g_{GABA}^{Tonic}$ and $g_{KCC2}$, $E_{GABA}$ can vary over ranges that correspond to excitatory, inhibitory and shunting effects of the resulting GABAergic current even for relatively small $g_{GABA}^{Tonic}$.

Next, we investigated the effect of simulated somatic GABAergic projections from the GPe on the firing rate of the model SNr neuron. This was achieved by simulating optogenetic stimulation of the model's somatic synapses at 40 Hz for 1 s. Four distinct types of SNr firing rate responses were

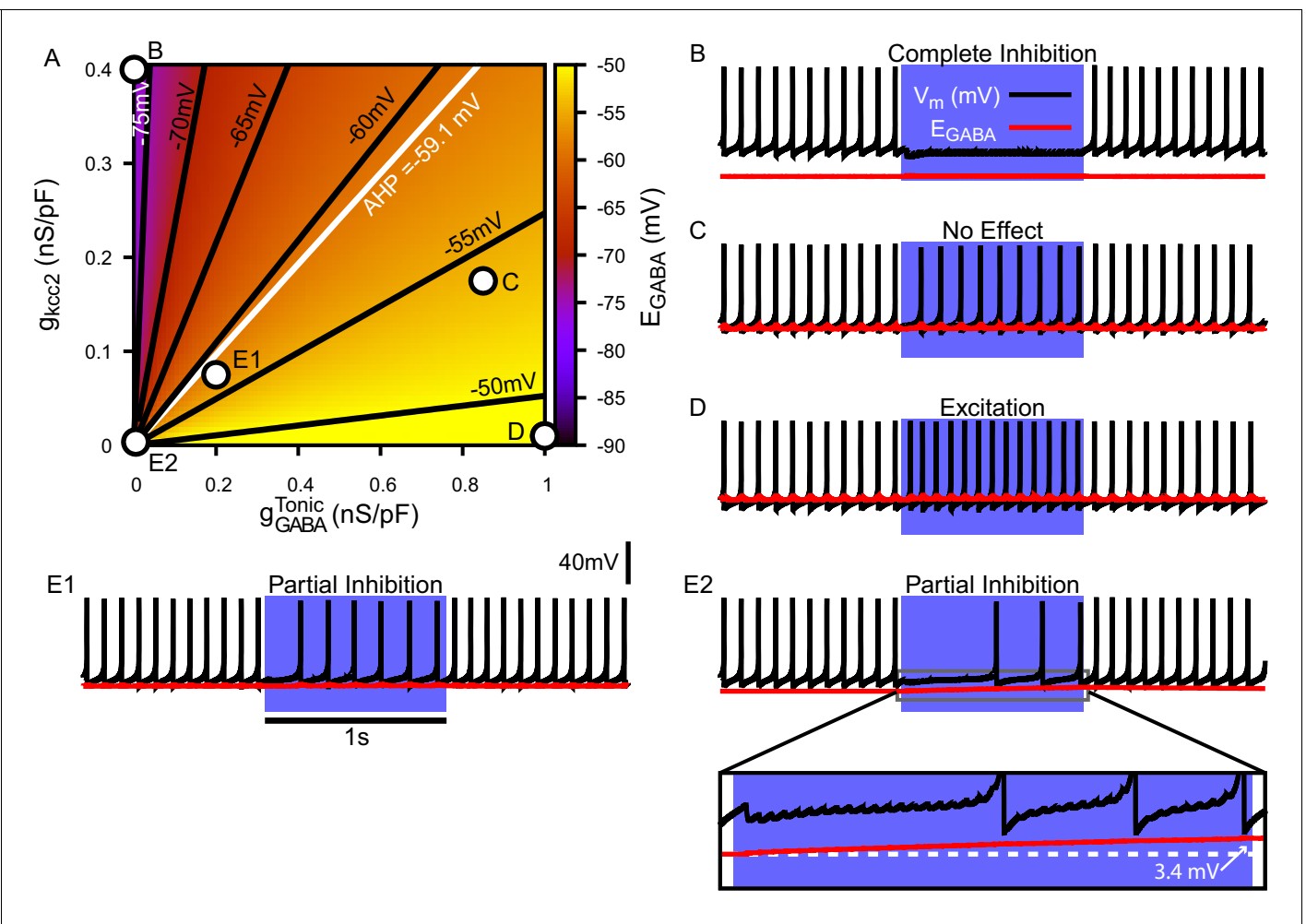

**Figure 4.** Tonic chloride conductance and extrusion capacity determine somatic $E_{GABA}$ and SNr responses to simulated 40 Hz GPe stimulation. (A) Dependence of somatic $E_{GABA}$ on the tonic chloride conductance ($g_{GABA}^{Tonic}$) and the potassium-chloride co-transporter KCC2 extrusion capacity ($g_{KCC2}$). (B–E) Examples of SNr responses to simulated indirect pathway stimulation at different positions in the 2D ($g_{GABA}^{Tonic}$, $g_{KCC2}$) parameter space, as labeled in panel (A). (E1 and E2) Notice the two distinct types of partial inhibition. Inset highlights the drift in $E_{GABA}$ during stimulation.

The online version of this article includes the following figure supplement(s) for figure 4:

**Figure supplement 1.** Biphasic SNr response to longer simulated GPe stimulation.

observed: 'complete inhibition', 'no effect', 'excitation', and 'partial inhibition' (*Figure 4B–E*). Additionally, two sub-types of partial inhibition occurred: (1) deletion of one or a few spikes followed by a step reduction in firing rate (*Figure 4E1*) and (2) complete inhibition followed by a late escape and continuation of spiking (*Figure 4E2*). The type of response in the model depends on the magnitude of $E_{GABA}$ relative to $V_m$ at the start of the stimulation, and, in the case of the second type of partial inhibition, the slow depolarizing drift of $E_{GABA}$ that is the result of intracellular $Cl^-$ accumulation. The effects of the short-term synaptic depression at these synapses on most of the SNr responses turns out to be minimal. This lack of effect arises because these synapses reach steady-state level of depression after approximately five stimulus pulses, which occurs after just 125.0 ms when stimulating at 40 Hz. The one exception occurs with the first type of partial inhibition, for which $g_{GABA}$ is large enough at the start of the stimulation window to cause an early spike deletion, after which depression can allow the reduced-rate firing to emerge.

We next performed a parallel analysis of the effects of simulated optogenetic stimulation of Str GABAergic projections in the dendritic compartment of the SNr model under the same stimulation protocol. As with the somatic compartment, we first characterized the relationship among $g_{GABA}^{Tonic}$, $g_{KCC2}$ and $E_{GABA}$ in the dendritic compartment and found that $E_{GABA}$ varies over a comparable range

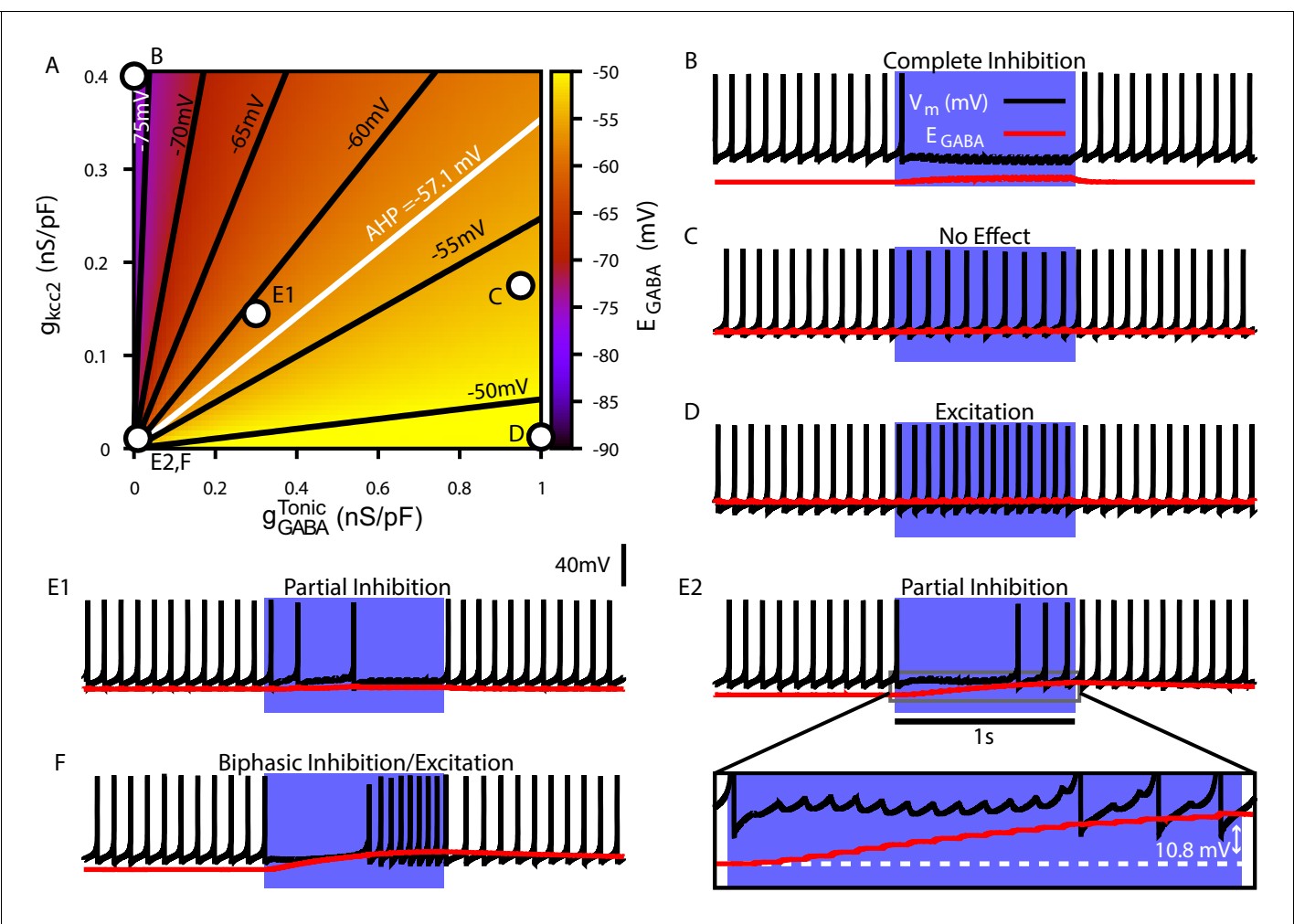

**Figure 5.** Tonic chloride conductance and extrusion capacity determine dendritic $E_{GABA}$ and SNr responses to 20 Hz Str stimulation. (A) Dependence of somatic $E_{GABA}$ on the tonic chloride conductance ($g_{GABA}^{Tonic}$) and the potassium-chloride co-transporter KCC2 extrusion capacity ($g_{KCC2}$). (B–F) Examples of SNr responses to simulated indirect pathway stimulation at different locations in the 2D ($g_{GABA}^{Tonic}, g_{KCC2}$) parameter space. $g_{GABA}^{Tonic}$ and $g_{KCC2}$ for each example are indicated in panel (A). (E1 and E2) Notice the two distinct types of partial inhibition. Inset highlights the drift in $E_{GABA}$ during stimulation. (F) Example of a biphasic inhibition-to-excitation response elicited by increasing the stimulation frequency to 40 Hz under the same conditions shown in E2. Alternatively, same response could be elicited by increasing the synaptic weight ($W_{GABA}^{Str}$). .

(−80 mV to −45 mV), depending on the balance of $Cl^-$ influx and extrusion rates (*Figure 5A*). Stimulation of the dendritic GABAergic synapses resulted in the same four response types seen in the somatic compartment with an additional 'biphasic inhibitory-to-excitatory' response and a slightly different pair of 'partial inhibition' responses (*Figure 5B–F*), one mediated by the short-term facilitation of direct pathway synapses. Specifically, with repeated stimulation, the strengthening of these synapses can induce a gradual slowing in the SNr firing rate throughout the simulation, which may eventually stop neuronal spiking (*Figure 5E1*). Despite this facilitation, a form of partial inhibition consisting of an initial pause in SNr spiking followed by a recovery of spiking can also occur in the model with direct pathway stimulation, mediated by a sufficiently large $Cl^-$ accumulation to allow the effects of dynamic $E_{GABA}$ to dominate the post-synaptic response (*Figure 5E2*). The biphasic inhibitory-to-excitatory response type is an extreme case of the partial inhibition shown in *Figure 5E2*. This biphasic response occurs when $E_{GABA}$ is initially hyperpolarized relative to $V_m$, the Str GABAergic conductance is strong and the $Cl^-$ extrusion capacity is weak, which allows for unusually rapid $Cl^-$ accumulation and subsequent depolarization of $E_{GABA}$ near or above the action potential threshold. The biphasic response type is more likely to occur with Str rather than GPe stimulation, due to the larger surface area-to-volume ratio and concomitant increased susceptibility to $Cl^-$ accumulation in the dendritic compartment that Str inputs target, relative to the soma. In our model biphasic responses to GPe inputs can be elicited under some conditions, see *Figure 4—figure supplement 1*.

## Optogenetic stimulation of GPe and Str GABAergic synaptic terminals in the SNr results in diverse neuronal responses

Our simulations in the previous sections predict that GABAergic inputs from the GPe and Str may produce a diverse range of effects on SNr activity depending on $E_{GABA}$ and $[Cl^-]_i$ levels and dynamics. To test these predictions, we optogenetically stimulated the synaptic terminals from D1 striatal neurons of the direct pathway and from GPe neurons of the indirect pathway in the SNr for 10 s periods. During stimulation, we performed patch clamp recordings of SNr activity. Experiments were conducted in in vitro slice preparations and patch clamp recordings were performed in cell attached mode to avoid perturbing the intracellular $Cl^-$ concentration critical for GABAergic signaling. In response to optogenetic stimulation, we found a wide array of SNr response types, which we classified into five categories: (1) complete inhibition - cessation of spiking; (2) partial inhibition - sufficient reduction of firing rate with or without a pause; (3) no effect - no change in firing rate; (4) excitation - sufficient increase in firing rate; and (5) biphasic - decrease or pause in spiking followed by an increase in firing rate above baseline. This heterogeneity in SNr responses may relate to differences in slicing-induced damage and corresponding baseline $E_{GABA}$ values or to other local factors. Example traces for the response types observed with GPe and Str stimulation are shown in *Figure 6 A1 and B1*, and the frequencies of occurrence for these responses are quantified in *Figure 6 A2 and B2*; see also *Figure 6—figure supplement 1* and *Figure 6—figure supplement 2* for raster plots and firing rate time courses for all stimulation frequencies tested. All response types could be induced by optogenetic stimulation of the GPe or the Str projection; however, with GPe stimulation, biphasic responses were slower to emerge (see *Figure 6—figure supplement 1*) and less common overall than with Str stimulation, consistent with the absence of biphasic responses in our 1 s simulations of GPe inputs and with slower $Cl^-$ accumulation, over several seconds, in the soma than in the dendrite. Biphasic responses do emerge with simulation of longer GPe stimulation in our model, see *Figure 4—figure supplement 1*. In a portion of the neurons partially inhibited by GPe or Str stimulation, the duration of the pause in spiking is longer than can be explained by short-term synaptic dynamics. Additionally, the number of partially inhibited neurons with a 'long pause' increases with stimulation frequency (GPe: 10 Hz, 1/25; 20 Hz, 8/26; 40 Hz, 13/29; 60 Hz, 16/24; Str: 10 Hz, 1/25; 20 Hz, 8/26; 40 Hz, 13/29; 60 Hz, 16/24). These findings, in addition to the observation of biphasic responses, are consistent with gradual $Cl^-$ accumulation and depolarization of $E_{GABA}$ during the stimulation period.

Previous computational modeling studies that have shown that, due to the larger surface area-to-volume ratio of dendrites relative to the soma, $Cl^-$ accumulation and depolarization of $E_{GABA}$ is faster in dendritic compared to somatic compartments (*Doyon et al., 2011*; *Ratté and Prescott, 2011*), and this result could explain why biphasic responses were almost never seen with GPe stimulation below 60 Hz. Nonetheless, $Cl^-$ accumulation and depolarization of $E_{GABA}$ may still arise, on a slower

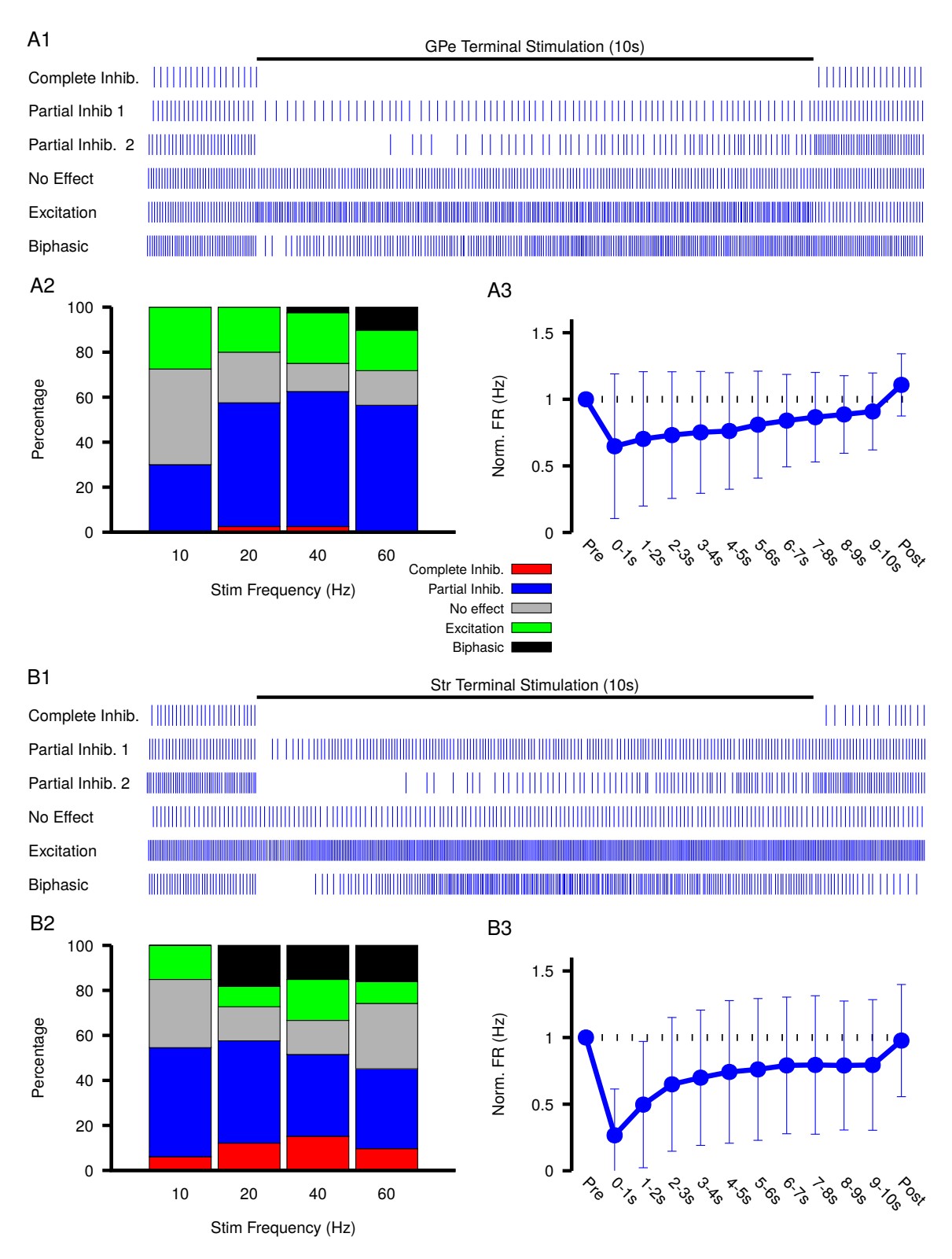

**Figure 6.** Characterization of experimentally observed SNr responses to optogenetic stimulation of (top) GPe and (bottom) Str projections to SNr in vitro. (**A1** and **B1**) Examples of response types observed for 10 s stimulation of GPe or Str projections. (**A2** and **B2**) Quantification types of SNr response to optogenetic stimulation at varying frequencies (GPe: $n = 4$ animals, 12 slices, 10 Hz, 20 Hz, and 40 Hz = 40 cells, 60 Hz = 39 cells; Str: $n = 4$ animals, 12 slices, 10 Hz, 20 Hz, and 40 Hz = 33 cells, 60 Hz = 31 cells). (**A3** and **B3**) Effect of GPe or Str stimulation on the firing rate of SNr neurons averaged

*Figure 6 continued on next page*

*Figure 6 continued*

across all trials for stimulation at 40 Hz. Error bars indicate the standard deviation. The 10 s stimulation period was broken into 1 s intervals to show the gradual weakening of inhibition during stimulation.

The online version of this article includes the following figure supplement(s) for figure 6:

**Figure supplement 1.** Summary of SNr responses to optogenetic stimulation of GPe synaptic terminals.

**Figure supplement 2.** Summary of SNr responses to optogenetic stimulation of Str synaptic terminals.

time scale, with stimulation of the indirect pathway. If slow $Cl^-$ accumulation and depolarization of $E_{GABA}$ are indeed occurring, then the strength of inhibition should slowly weaken during stimulation, which will result is a slow increase in firing rate during the stimulation period.

Measurements of spiking frequency relative to baseline during and after stimulation of the GPe and Str projections as a function of stimulation frequency (*Figure 6 A3 and B3*) support the idea that $E_{GABA}$ dynamics may contribute to synaptic integration within the SNr. For this analysis, we divided the stimulation period into 1 s intervals in order to assess any dynamic changes in the strength of the input over the course of stimulation. We found that Str projections are initially more effective at inhibiting SNr spiking relative to GPe projections (Str: 72.3–76.7% peak reduction, GPe: 43.1–61.9% peak reduction). Interestingly, for both GPe and Str projections, the strength of inhibition decreases on average during the stimulation period, consistent with slow accumulation of intracellular chloride. Moreover, the loss of firing rate reduction was most prominent for Str stimulation at high frequency, despite short-term synaptic facilitation known to occur at these synapses (*Connelly et al., 2010*; *Lavian and Korngreen, 2016*), consistent with the emergence of some excitatory and biphasic SNr responses in that regime.

The diversity of experimental responses to GPe and Str stimulation seen in *Figure 6* support the idea that GABAergic synaptic transmission in the SNr is not purely inhibitory and may even be excitatory in some neurons. In the following sections we return to our computational model to explore the functional significance of this finding in physiologically relevant settings.

## $E_{GABA}$ tunes local SNr synchrony and may promote slow oscillations

In addition to receiving GABAergic projections from the GPe and Str, SNr neurons interact locally through GABA$_A$-mediated synaptic transmission (*Mailly et al., 2003*; *Brown et al., 2014*; *Higgs and Wilson, 2016*). The role of these synapses is unclear; however, they have been proposed to regulate synchronization of SNr activity (*Higgs and Wilson, 2016*). Levels of $E_{GABA}$ will affect the strength and polarity (inhibitory, shunting, excitatory) of these interactions. Therefore, we next used our computational model to characterize how variations in $E_{GABA}$, potentially due to differences in GPe firing rates, affect these local SNr interactions.

On average, a given SNr neuron receives GABAergic synaptic projections from 1 to 4 neighboring SNr neurons (*Higgs and Wilson, 2016*). Consequently, synaptic interactions between SNr neurons result in brief synaptic transients that have been proposed to impact neuronal synchrony incrementally by changing the oscillatory phase of the post-synaptic neuron. Therefore, we first characterized how transient GABAergic stimulation modulates the phase of our model SNr neuron as a function of the phase of the SNr oscillation at which the stimulation occurs, using phase response curves (PRCs) (*Ermentrout, 1996*; *Ermentrout and Terman, 2010*) computed for an array of values of $E_{GABA}$ (see *Figure 7*). The PRCs that we obtained for hyperpolarized values of $E_{GABA}$ are qualitatively consistent with those found previously for mouse SNr neurons in brain slices with $E_{GABA} \approx -65mV$ (*Simmons et al., 2018*).

PRCs can be used to predict the synchrony between two oscillating neurons that interact synaptically (*Ermentrout, 1996*; *Jeong and Gutkin, 2007*; *Ermentrout and Terman, 2010*; *Smeal et al., 2010*). We applied this idea with our computationally-generated PRCs to predict the synchrony in a network of two SNr neurons under two configurations, unidirectional and bidirectional synaptic connectivity (*Figure 8*). For the unidirectional case a first, presynaptic SNr neuron stimulates a second, postsynaptic one. Phases of the presynaptic neuron's ongoing oscillation at which the firing of the postsynaptic neuron will become locked can be predicted by finding locations where the PRC crosses the horizontal (phase) axis. Although all crossings represent fixed points and hence phases at which locking can theoretically occur, only those with a positive slope are stable and are predicted

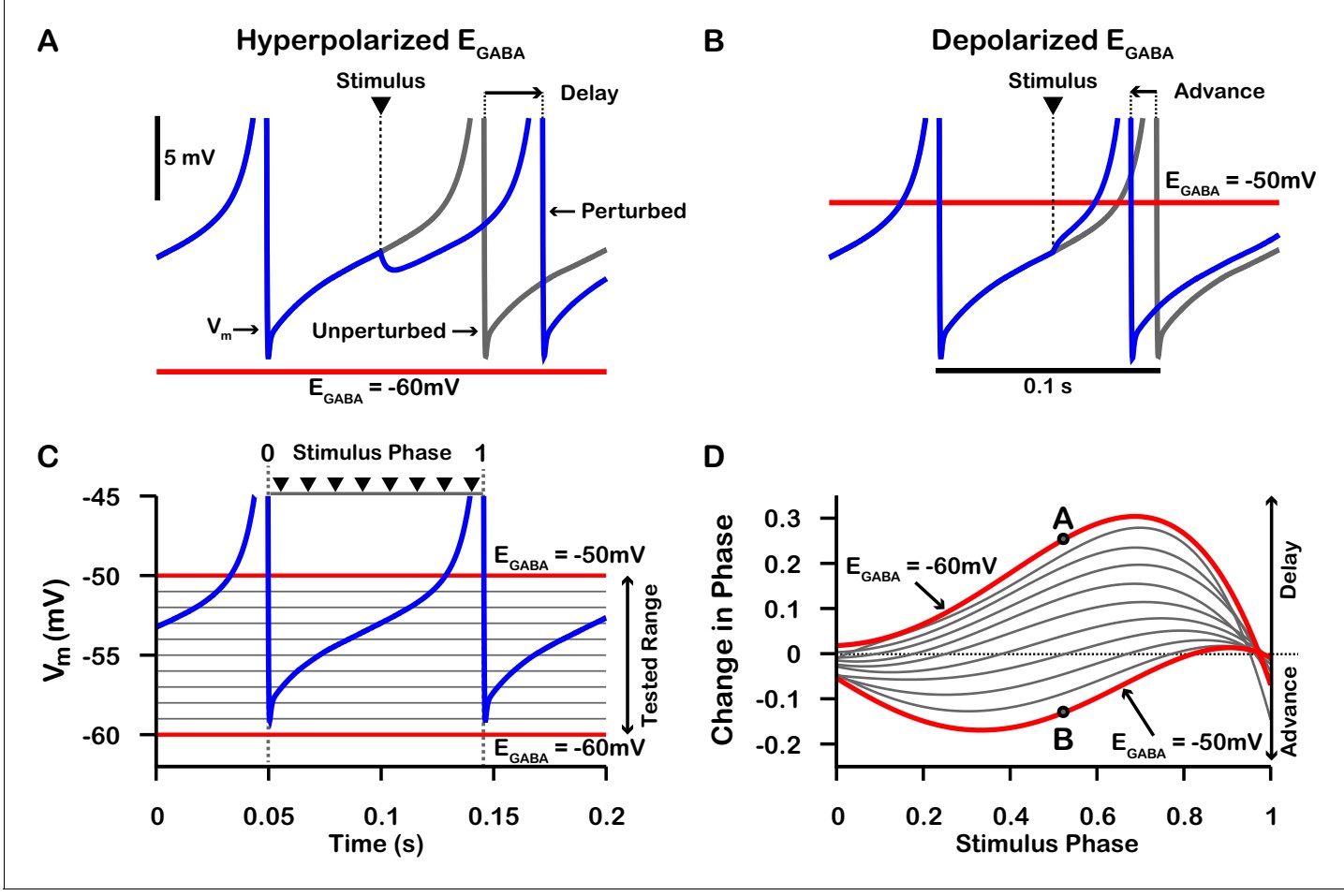

**Figure 7.** Phase response curves (PRCs) of the model SNr neuron depend on $E_{GABA}$. (**A** and **B**) Example traces illustrating the effect of a single GABAergic synaptic input on the phase of spiking in a simulated SNr neuron for hyperpolarized and depolarized $E_{GABA}$, respectively. (**C**) For an ongoing voltage oscillation of a spiking SNr neuron (blue trace), we define a phase variable as progressing from 0 immediately after a spike to one at the peak of a spike. As $E_{GABA}$ is varied from –60 mV to –50 mV, progressively more of the SNr voltage trace lies below $E_{GABA}$, where GABAergic inputs have depolarizing effects. (**D**) PRCs computed for a model SNr neuron in response to GABAergic input stimuli arriving at different phases of an ongoing SNr oscillation. As $E_{GABA}$ is varied from –60 mV to –50 mV, the PRC transitions from a curve showing a delay of the next spike for most stimulus arrival phases, through some biphasic regimes, to a curve showing an advance of the next spike for almost all possible phases. In panel D, the A and B labels at approximately 0.5 phase on the $E_{GABA} = -60\,mV$ and $E_{GABA} = -50\,mV$ PRCs correspond to the examples shown in panels A and B. The conductance of the synaptic input was fixed at 0.1 nS/pF in order to produce deflections in $V_m$ for hyperpolarized $E_{GABA}$ that are consistent with data presented in *Higgs and Wilson, 2016*.

to arise robustly and be observed in simulations (e.g., *Ermentrout, 1996*; *Ermentrout and Terman, 2010*). By tracking the fixed points, we found that in the unidirectional case, the locked phase relation between the two SNr neurons is predicted to go from synchrony, or phase 0, to progressively more asynchronous phase locking and then back toward synchrony again as $E_{GABA}$ depolarizes from —60 mV to –50 mV, with perfectly anti-phase spiking for $E_{GABA} \approx -53\,mV$ (*Figure 8A4*). We also observe that phase locking is predicted to be unstable (indicated by open circles) for sufficiently negative $E_{GABA}$ (less than $\approx$ –57 mV).

To test these predictions computationally, we simulated the unidirectionally connected two-neuron network and recorded the timing of synaptic inputs in the phase of neuron 2 (Input Phase). We found that the predicted synchrony/asynchrony is in good agreement with our simulations, and is indicated by the distributions of the input phase histograms shown in *Figure 8A4* (gray curves). Interestingly, for relatively hyperpolarized $E_{GABA}$ where synchronous phase locking is predicted to be unstable we observe that, instead of phase locking, slow oscillations in the phase of the postsynaptic neuron relative to that of presynaptic neuron begin to emerge. Correspondingly, the distribution of

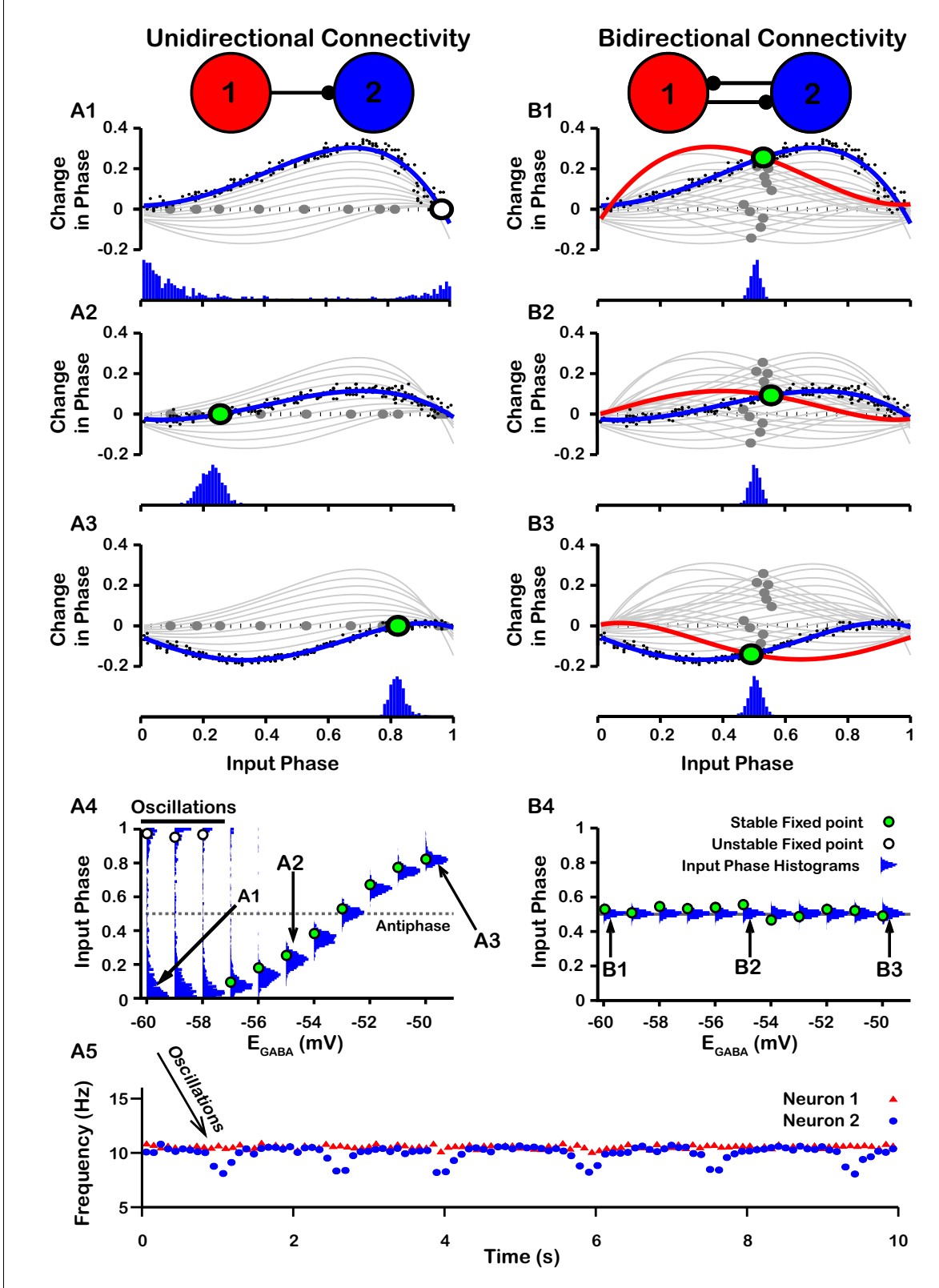

**Figure 8.** Effect of $E_{GABA}$ on SNr synchrony in a unidirectional (left) and bidirectional (right) synaptically connected two-neuron network. (A1-A3 and B1-B3) (Top) Identification of PRC fixed points and (Bottom) histogram of the timing of synaptic inputs in the phase of neuron 2 (Input Phase) as a function of $E_{GABA}$. Recall that positive changes in phase correspond to delays. (A1,B1) $E_{GABA} = -60\,mV$; (A2,B2) $E_{GABA} = -55\,mV$; (A3,B3) $E_{GABA} = -50\,mV$. Black dots indicate dataset used to generate PRC in red/blue. Stable and unstable fixed points are indicated by green and white filled circles, respectively. *Figure 8 continued on next page*

*Figure 8 continued*

For reference, all PRCs and fixed points are included in gray for all values of $E_{GABA}$ tested. (**A4,B4**) Effect of $E_{GABA}$ on SNr phase locking. Blue histograms show the distribution of synaptic inputs relative to the phase of neuron 2 (input phase) for the two network simulations for different levels of $E_{GABA}$. Green and white filled circles indicate the stable and unstable locking predicted by analysis of PRCs. Note the unstable fixed points for the lowest values of $E_{GABA}$ in the unidirectional case. (**A5**) In the unidirectional case, slow 1 Hz oscillations in the frequency of neuron 2 arise due to phase slipping at hyperpolarized values of $E_{GABA}$.

The online version of this article includes the following figure supplement(s) for figure 8:

**Figure supplement 1.** Schematic illustration of the convergence toward anti-phase locking in a birectionally coupled pair of SNr neurons.

presynaptic neuron phases when the postsynaptic neuron fires spreads out across the [0,1] interval and the frequency of firing of neuron the postsynaptic neuron repeatedly drifts below that of the presynaptic neuron, at a rate of about 1 Hz (*Figure 8A5*). The mechanism underlying these slow oscillations will be discussed in more detail below.

For bidirectional connectivity, we no longer have a clear distinction between a pre- and a postsynaptic neuron, and instead we just refer to neuron 1 and neuron 2. Due to the symmetry of the network, we can plot the PRC for neuron 1 together with that of neuron 2 by reflecting the PRC for neuron 2 about the mid-point of the phase axis, 0.5 (see *Materials and methods* for more detail). Phase locking between the two neurons can then be predicted by finding the intersections (fixed points) of these two PRCs (*Figure 8B1–B3*). By symmetry, a value near 0.5 is always a fixed point in this case, and we found that this was the only fixed point for the bidirectional system and remained stable regardless of the value of $E_{GABA}$ (*Figure 8B1-4*); see also *Figure 8—figure supplement 1* for a schematic illustration of how this anti-phase locking develops. Again, this prediction was tested by simulating the bidirectionally connected two-neuron network and recording the input phase in neuron 2. The predicted asynchrony between the two neurons is in good agreement with our simulations, in which the phase relationship between the two neurons remained tightly distributed around 0.5 for all values of $E_{GABA}$ tested (*Figure 8B4*).

The slow oscillations of approximately 1 Hz seen with unidirectional connectivity can be understood by taking a closer look at the PRCs calculated for $E_{GABA}$ less than approximately –57 mV in the unidirectional case (*Figure 8A*). For these values of $E_{GABA}$, the PRCs only have unstable fixed points. Under these conditions, the phase of neuron 2 relative to neuron 1 is delayed by different amounts across successive inputs from neuron 2 (or possibly advanced if inputs arrive during a specific narrow phase window). Moreover, based on the shape of the PRC, the magnitude of change in phase is large when phase is away from 0 and 1, such that spiking is asynchronous, and small when the phase of is nearly synchronous. As a result, the network remains close to synchrony most of the time but with approximately periodic asynchronous excursions, a phenomenon referred to as phase slipping (*Thounaojam et al., 2014*). We refer to the oscillations that arise through phase slipping as PS oscillations. The frequency of phase slipping is determined by the number of stimulus kicks needed for the phase to progress through one full cycle, which in turn is determined by the shape of the PRC. For example, one full phase slipping cycle is illustrated in *Figure 9A–B* for $E_{GABA} = -60\,mV$. As previously mentioned, the slow oscillation in phase is also seen as a periodic negative excursion in the frequency of spiking (*Figure 8A5*).

Since the frequency of PS oscillations is determined by the shape of the PRC and the PRC is in part determined both by $E_{GABA}$ and by the weight/conductance of the synaptic projection from the other neuron ($W_{GABA}^{SNr}$), changes in $E_{GABA}$ or $W_{GABA}^{SNr}$ should affect the phase slipping frequency. Therefore, we also characterized the relationship between $E_{GABA}$ and the frequency of the phase slipping for different values of $W_{GABA}^{SNr}$. In our simulations, we found that phase slipping oscillations begin at approximately $E_{GABA} = -56\,mV$ and linearly increase in frequency as $E_{GABA}$ is held at progressively more hyperpolarized values (*Figure 9C*). The hyperpolarization of $E_{GABA}$ leads to stronger inhibition and hence a larger PRC amplitude, which allows for the postsynaptic neuron to progress through the full phase range on fewer cycles (i.e., at a higher frequency). Moreover, the magnitude of the slope of the linear relationship between $E_{GABA}$ and frequency increases/decreases with increases/decreases in the strength of $W_{GABA}^{SNr}$ due to similar effects. We also simulated SNr neurons with different levels of applied current, leading to different firing rates, but this variability did not strongly impact resulting oscillation frequencies.

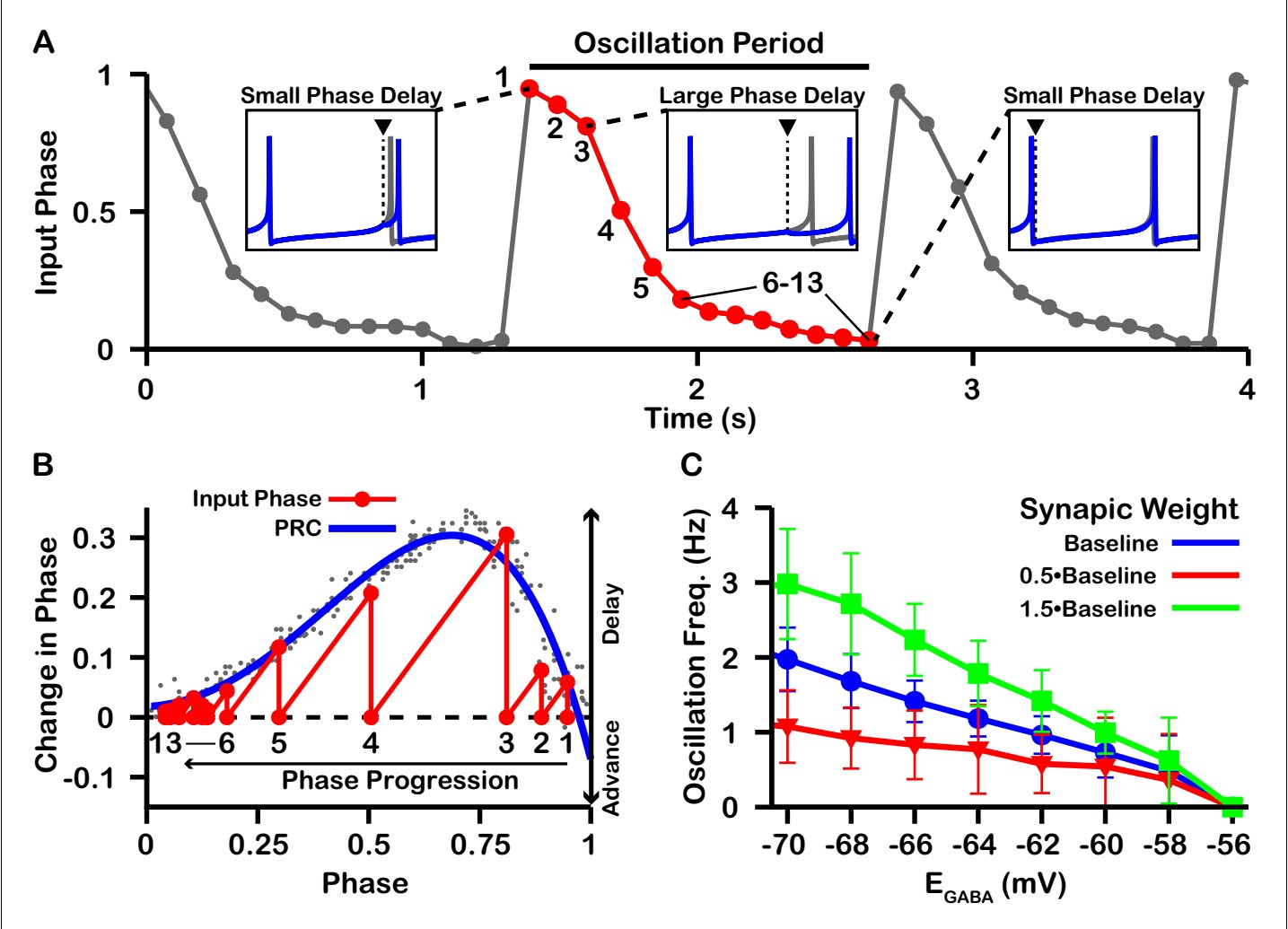

**Figure 9.** Characterization of phase slipping oscillations in the unidirectionally connected two-neuron network. (A) Illustration of the phase of the postsynaptic neuron at the moment when it receives each input from the presynaptic neuron (input phase) for the unidirectionally connected two neuron network as a function of time for $E_{GABA} = -60\,mV$. Dots denote phases of the postsynaptic neuron when the presynaptic neuron spikes. The phase value of 1 corresponds to the postsynaptic neuron being at spike threshold. Insets show the timing of the presynaptic neuron spike (black triangle/dashed line), the phase of the postsynaptic neuron spike after it receives the input (blue), and the spike train of the postsynaptic neuron in the absence of input (gray). The red cycle is used in B. (B) Overlay of the PRC generated for $E_{GABA} = -60\,mV$ and the resulting progression of phase for one full phase slipping oscillation. Light gray dots indicate the data points used to generate the blue PRC. Recall that positive changes in phase correspond to delays. (C) The frequency of phase slipping increases as $E_{GABA}$ decreases, with a steeper relationship for larger synaptic weight ($W_{GABA}^{SNr}$) between the SNr neurons. .

## Phase slipping and phase advancing oscillations

Next we investigated how changing the presynaptic neuron's firing rate affects the relationship between $E_{GABA}$ and SNr synchrony and the emergence of slow oscillations. Under normal conditions in in vitro slice preparations, SNr neurons have been measured to spike at 10.4 ± 0.2 Hz (*Zhou et al., 2008*) and 10.7 ± 0.9 Hz (*Zhou et al., 2009*). We increased/decreased the firing rate of the presynaptic neuron over a wider range, from below 10 Hz to above 11 Hz, via current injection (*Figure 10A*) and examined the resulting dynamics in the unidirectional network. These simulations show that the synchrony relationship between the neurons is maintained regardless of the presynaptic neuron's firing rate. For each choice of presynaptic firing rate, we observe a range of $E_{GABA}$ values supporting pure phase locking and another range supporting PS oscillations. The slower the presynaptic firing rate, the more hyperpolarized $E_{GABA}$ needs to be for PS oscillations to occur (*Figure 10C1-4*). Moreover, a new feature that appears when the presynaptic firing rate is slowed is

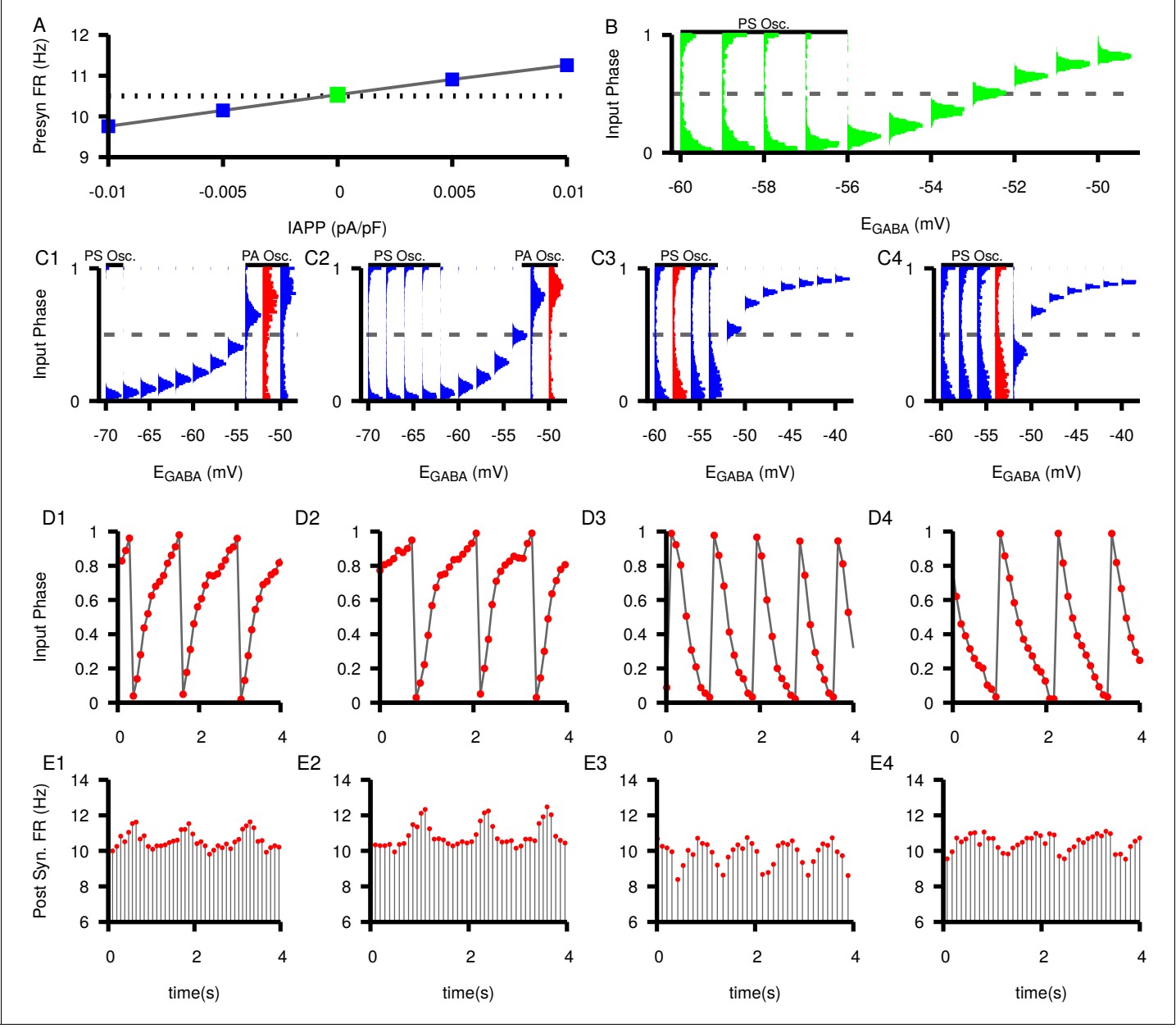

**Figure 10.** Effects of changing the presynaptic firing rate on synchrony and postsynaptic oscillations in a feed-forward SNr neuron pair. (A) Tuning curve for presynaptic firing rate (FR) versus applied current, $I_{APP}$. Dashed line indicates the baseline firing rate (10.5 Hz) with no applied current. (B) Histograms of the input phase in the postsynaptic neuron under baseline conditions ($I_{APP} = 0\,pA/pF$). (C1–C4) Input phase histograms for different presynaptic firing rates (9.76 Hz, 10.15 Hz, 10.91 Hz, 11.26 Hz from left to right). $E_{GABA}$ ranges are not the same in all panels. Regions of phase slipping (PS) and phase advancing (PA) oscillations are each indicated by a solid horizontal bar. Example oscillations in input phase (D1–D4) and postsynaptic firing rate (E1–E4) at specific values of $E_{GABA}$ for different presynaptic firing rates highlighted in red for the corresponding D panels. Notice that the PA oscillations in C1-2 and D1-2 result in periodic increases in the postsynaptic firing rate in E1-2 whereas PS oscillations in C3-4 and D3-4 result in periodic decreases in firing rate in E3-4.

The online version of this article includes the following figure supplement(s) for figure 10:

**Figure supplement 1.** The relationship between $E_{GABA}$ and phase locking and the emergence of slow oscillations are maintained at least up to in vivo SNr firing rates.

**Figure supplement 2.** Effect of increasing noise on SNr phase relationships as a function of $E_{GABA}$.

**Figure supplement 3.** Effect of synaptic delay on the relationship between $E_{GABA}$ and presynaptic/postsynaptic phase locking.

**Figure supplement 4.** Relationship between postsynaptic firing properties as a function of $E_{GABA}$ and varying degrees of synchrony between two presynaptic neurons.

*Figure 10 continued on next page*

*Figure 10 continued*

**Figure supplement 5.** Relationship between postsynaptic firing properties as a function of $E_{GABA}$ and varying degrees of synchrony between three presynaptic neurons.

**Figure supplement 6.** Relationship between postsynaptic firing properties as a function of $E_{GABA}$ and varying degrees of synchrony between four presynaptic neurons.

**Figure supplement 7.** Effects of varying $E_{GABA}$ on post synaptic dynamics in the three-neuron motif where neuron 1 projects to neuron 2 and neuron 3 and neuron 2 projects to neuron 3 (motif number 10 from *Song et al., 2005*).

a second type of oscillations, which we term phase advancing (PA) oscillations. These arise when $E_{GABA}$ is relatively depolarized (*Figure 10C1-2*) and manifest as transient increases in the postsynaptic neuron's firing rate, as shown in *Figure 10D1,D2,E1,E2* and contrasting with the PS oscillations shown in *Figure 10D3,D4,E3,E4*.

## Robustness

We also systematically tested the robustness of the unidirectionally connected two-neuron network oscillations and phase locking predictions to several factors: (1) increased SNr neuron firing rates, (2) the presence of noise, (3) synaptic delays, and (4) the number of presynaptic neurons projecting to each postsynaptic target; see *Figure 10—figure supplements 1–6*. Robustness results were similar for mutually connected pairs. For (4), we consider (a) various numbers of presynaptic neurons projecting to a single postsynaptic cell (*Figure 10—figure supplements 4–6*) and (b) activity patterns within a specific three-cell motif (*Figure 10—figure supplement 7*). In general, we found that the predicted oscillations and phase locking as a function of $E_{GABA}$ in the model are extremely robust to these factors.

Finally, we simulated a network of 100 model SNr neurons each receiving synaptic inputs from between 0 and 8 other SNr neurons (*Figure 11A*). The heterogeneity in inputs led to variability across individual neurons' mean firing rates (*Figure 11B*) and spike rate CV (*Figure 11C*) at hyperpolarized $E_{GABA}$, with more uniform spiking at more depolarized $E_{GABA}$ (*Figure 11D*). Interestingly, strong oscillations in the 0–4 Hz frequency range were prevalent in many cells within the network at relatively hyperpolarized and relatively depolarized $E_{GABA}$ (*Figure 11E*), consistent with the smaller circuit results and with the identified framework of PS and PA oscillations. Examples of the power spectra and firing rate time courses for oscillating neurons at $E_{GABA} = -60\,mV$ and $E_{GABA} = -50\,mV$ are shown in *Figure 11F*; the upward and downward deviations from baseline ($\approx$ 10 Hz) in these plots support the suggestion that PS and PA oscillations persist in the larger SNr network. Overall, these results represent a strong indication of the robustness of our findings.

## Optogenetic stimulation of GPe neurons suppresses SNr oscillations

Slow oscillations have been reported in the SNr in vivo under dopamine depleted (DD) conditions in lightly anaesthetized (*Walters et al., 2007*) and awake behaving animals (*Whalen et al., 2020*). Our simulations predict that similar slow oscillations will occur when $E_{GABA}$ is equal to or hyperpolarized relative to the membrane AHP. Assuming that these oscillations are driven by the mechanism described in *Figure 9*, manipulations that depolarize $E_{GABA}$ should reduce and stop such oscillations. As illustrated in *Figure 4*, changing the tonic $Cl^-$ conductance to the soma is one way to depolarize $E_{GABA}$. This could be achieved by increasing the firing rate of GPe neurons. Therefore, next we examined if these slow oscillations are suppressed by optogenetic stimulation of GPe neurons in the SNr. Consistent with previous descriptions (*Walters et al., 2007*; *Whalen et al., 2020*), under DD conditions we found slow oscillations in the firing rates of SNr neurons (*Figure 12A–C*). The frequency of the oscillations was characterized by finding the peak in the power spectral density (PSD) as described in *Whalen et al., 2020* and shown in *Figure 12B*. We identified five oscillatory units with frequencies ranging from 1.46 Hz to 1.95 Hz (mean ± SD = 1.7 ± 0.204 Hz, *Figure 12C*). In these units, optogenetic stimulation of GPe terminals in the SNr had limited effect on SNr firing rates during a 30 s stimulation period (*Figure 12D*; see *Materials and methods* for a full description of the experimental preparation and stimulation protocol). Yet, stimulation of GPe terminals in the SNr significantly reduced the power in the PSD in the 0.25–4.0 Hz band (*Figure 12E*). The impact of GPe stimulation on oscillations but not firing rate in the SNr is consistent with our simulations and

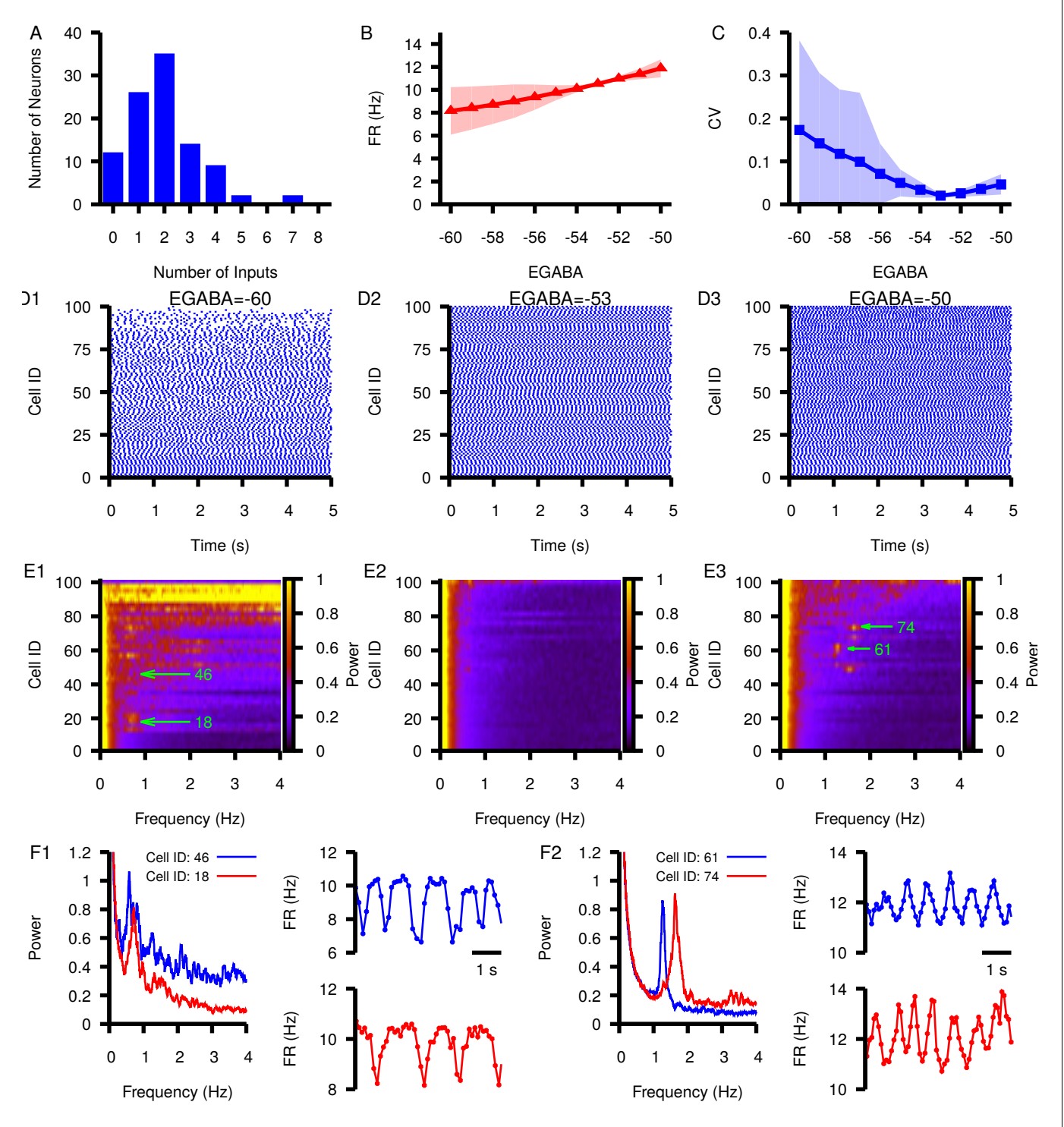

**Figure 11.** Effect of varying $E_{GABA}$ in a network of 100 model SNr neurons with random, sparse connectivity. (A) Histogram showing the number of neurons receiving zero to eight synaptic inputs. (B) Mean network firing rate as a function of $E_{GABA}$. (C) Mean network CV as a function of $E_{GABA}$. Shaded regions in B and C represent standard deviation. (D1–D3) Example raster plots of spikes in the network for three different values of $E_{GABA}$. (E1–E3) Power spectrum for each neuron in the network for the same three values of $E_{GABA}$ used in (D1–D3). Rows in D and E panels are sorted by the number of inputs from least (cell 1) to most (cell 100). Green arrows point out peaks in the power spectra of example neurons examined in the following panels. (F1 and F2) Left panel: Power spectrum for two example neurons with peaks indicating slow oscillations. Right panels: slow oscillation in the instantaneous firing rate in the two example neurons. These are shown for $E_{GABA} = -60\,mV$ (F1) and for $E_{GABA} = -50\,mV$ (F2).

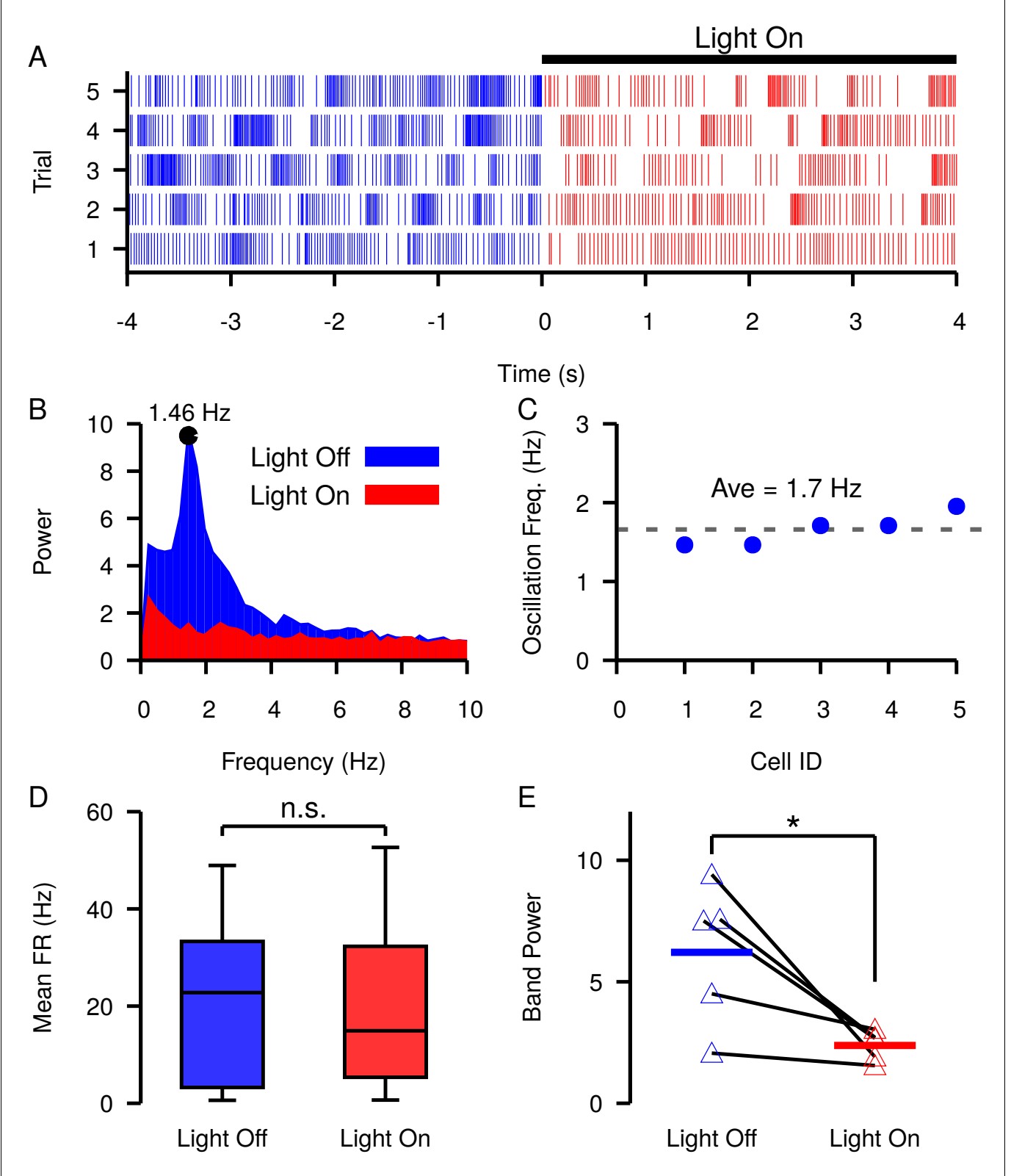

**Figure 12.** Slow oscillations in the SNr seen under dopamine depleted conditions in vivo are suppressed by channelrhodopsin-2 optogenetic stimulation of GABAergic GPe terminals in the SNr. (A–B) Example (A) raster plot and (B) power spectrum of a single spiking unit in SNr without (blue) and with (red) optogenetic stimulation of GPe terminals over multiple trials. (C) Frequencies of slow oscillations in the 12 unit dataset before optogenetic stimulation. (D) Distribution of single unit firing rates without (blue) and with (red) optogenetic stimulation for all recorded units (n = 12). *Figure 12 continued on next page*

*Figure 12 continued*

Notice that stimulation has no significant effect on firing rate (t-test p=0.8531). (**E**) Band power (0.75–3.0 Hz) without (blue) and with (red) optogenetic stimulation for oscillatory units (n = 5). Solid blue and red horizontal bars indicate mean band power. Notice that stimulation significantly reduces the power of the slow oscillations (t-test p=0.0341). Recordings were collected from four animals. .

suggests that slow oscillations in the SNr seen under DD conditions may be due to the phase slipping mechanism described in *Figure 9*. These data also suggest that a role of the GPe may be to tune SNr dynamics by modulating the tonic $Cl^-$ conductance and $E_{GABA}$ in the SNr.

## $E_{GABA}$ tunes the strength of direct pathway inhibition and may affect response times in perceptual decision-making tasks

In tasks involving perceptual decision-making, visual motor responses (saccades) are thought to be triggered when evidence accumulates above some threshold level. Experiments suggest that the BG is involved is regulating the dynamics of these visual motor responses (*Basso and Wurtz, 2002*; *Basso et al., 2005*; *Shires et al., 2010*; *Sato and Hikosaka, 2002*). In the BG, evidence accumulation is thought to be represented by a ramping increase in the firing rate in striatal neurons of the direct pathway (*Ding and Gold, 2010*) that, above some threshold, generates a pause in SNr activity (*Wei et al., 2015*; *Dunovan et al., 2019*). The pause in SNr spiking disinhibits downstream motor targets and allows the initiation of a selected action. As we have shown above, the effect(s) of striatal inputs on the firing rate and pattern of SNr neurons is highly dependent on $E_{GABA}$, which, in turn, is determined by the tonic chloride conductance and the $Cl^-$ extrusion capacity of the KCC2 pump. Therefore, changes in $E_{GABA}$ are predicted to modulate the threshold at which ramping striatal activity will generate a pause in SNr firing.

In the previous section we argued that the tonic GABAergic input from GPe neurons of the direct pathway may provide a mechanism to tune $E_{GABA}$ in the soma of SNr neurons. Assuming that the coupling between the somatic and dendritic compartments is sufficiently strong, the tonic somatic $Cl^-$ conductance provided by GPe inputs may also tune $E_{GABA}$ in the dendritic compartment. To illustrate this idea, we first constructed a population of 100 SNr neurons with a baseline firing rate turned up to ≈ 25 Hz in order to better represent in vivo conditions (*Freeze et al., 2013*; *Mastro et al., 2017*; *Willard et al., 2019*; *Figure 13A,B*). In this set of simulations $[Cl^-]_i$ in the somatic and dendritic compartments interact by the addition of a coupling term (see *Materials and methods* for a full description). Next, we characterized $E_{GABA}$ in the somatic and dendritic compartments as a function of the tonic somatic $Cl^-$ conductance (representing the tonic GABAergic GPe input). As expected, $E_{GABA}$ depolarizes in both compartments as the somatic chloride conductance is increased (*Figure 13C*). In the dendritic compartment in particular, $E_{GABA}$ ranges from just below –75 mV with no chloride conductance to approximately –57 mV with a 1.0 nS/pF $Cl^-$ conductance in the soma.

Finally, we characterized the relationship between tonic $Cl^-$ conductance and the time required to decrease the mean SNr population firing rate below thresholds of 1 Hz, 5 Hz, and 10 Hz in response to a ramping striatal input (*Figure 13D–F*). As the tonic $Cl^-$ conductance increases, $E_{GABA}$ becomes less hyperpolarizing (*Figure 5*) and hence more time is needed to push SNr activity below threshold; for high enough $Cl^-$ conductance, the ramping striatal input is unable to suppress SNr firing below 1 Hz. These simulations illustrate a plausible mechanism through which the tonic $Cl^-$ conductance provided by the level of GPe activity may be able to tune dendritic (and somatic) $E_{GABA}$, altering SNr responses to direct pathway striatal inputs and, ultimately, the response times in perceptual decision-making tasks. Qualitatively, these results are unchanged if the dendritic compartment is divided into multiple thin dendrites as opposed to one lumped dendrite, although this modification can hasten the rise and decay of $E_{GABA}$ after the onset and offset of a stimulus, respectively; see *Figure 13—figure supplements 1* and *2*.

## Discussion

In this work, we used computational modeling to explain and make predictions about the responses of SNr neurons to the streams of GABAergic input that they receive from the GPe and striatum (Str), as well as the effects of local interactions within the SNr. Results from previous experiments and

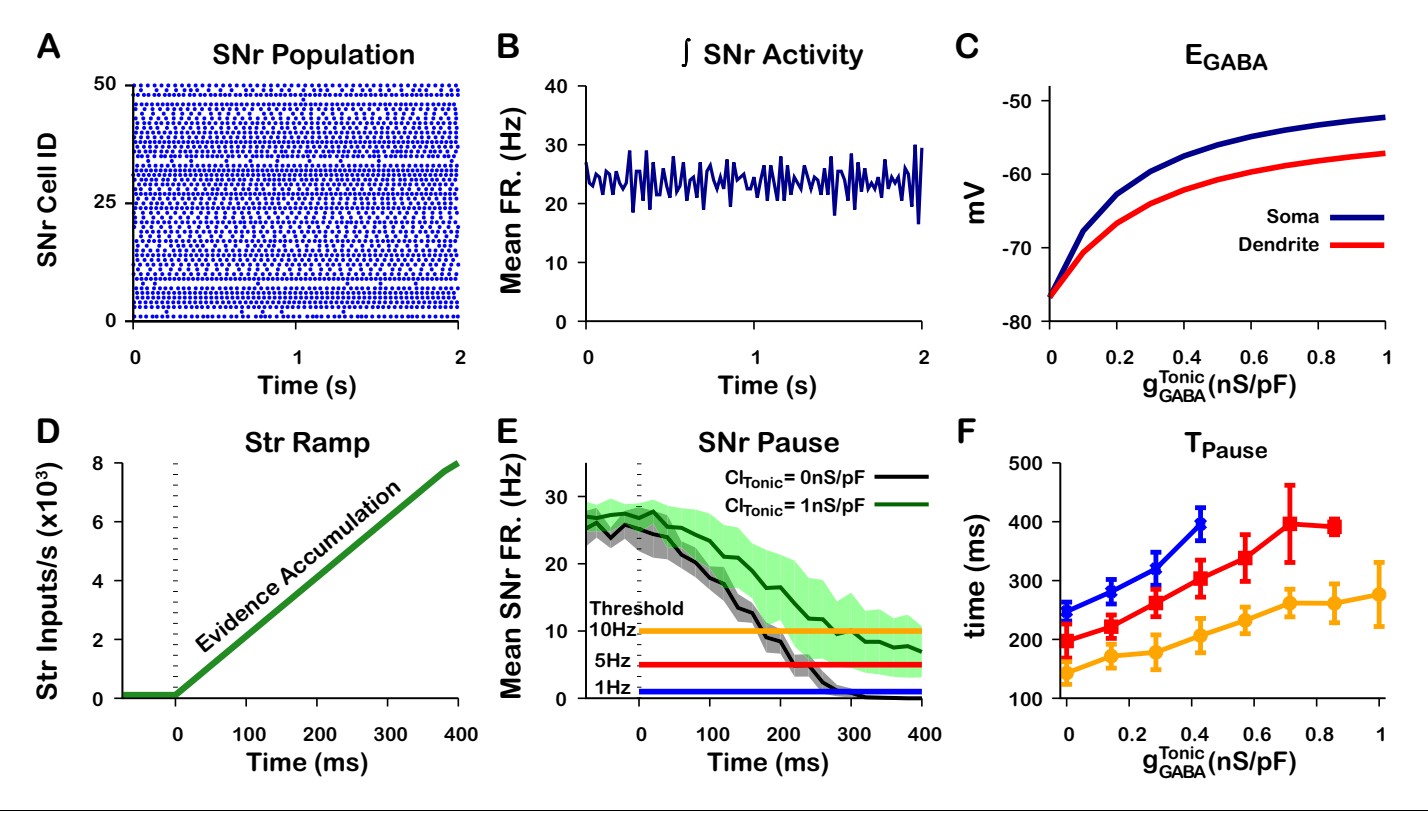

**Figure 13.** Tonic somatic $Cl^-$ conductance affects somatic and dendritic $E_{GABA}$ and tunes SNr responses to Str inputs. (**A**) Raster plot of spikes in the simulation of an SNr network model containing 50 simulated neurons that receive tonic somatic inhibition from GPe projections. (**B**) Integrated SNr population activity gives a mean firing rate of about 23 Hz, as seen in in vivo conditions (*Freeze et al., 2013*; *Mastro et al., 2017*; *Willard et al., 2019*). (**C**) Increasing tonic $Cl^-$ depolarizes somatic and dendritic $E_{GABA}$. (**D**) Ramping Str synaptic inputs used to represent evidence accumulation in a perceptual decision-making task. (**E**) Inhibition and pause generation in the SNr during evidence accumulation/ramping Str activity, for two different tonic somatic $Cl^-$ conductances. (**F**) Increasing the tonic $Cl^-$ conductance lengthens $T_{pause}$, the time for the SNr firing rate to drop below threshold (colors correspond to threshold levels in E). If the tonic conductance becomes too great, then SNr firing cannot be pushed to arbitrarily low rates. . The online version of this article includes the following figure supplement(s) for figure 13:

**Figure supplement 1.** Switching from a single dendrite to multiple thin dendrites increases rate but not magnitude of $Cl^-$ accumulation and subsequent depolarization of $E_{GABA}$ in response to simulated $40\,Hz$ Str stimulation.

**Figure supplement 2.** Increasing the number of dendrites has no qualitative effect on the the time it takes to generate a pause in SNr activity in response to ramping Str activity.

from those reported in this paper show that each of these channels, when activated on its own, can induce diverse patterns of SNr spiking. Our simulations show that these responses can result from varying levels of the GABA$_A$ reversal potential, short-term plasticity, and in some cases intracellular $Cl^-$ dynamics. GPe neurons, with somatic synapses on SNr neurons and relatively high sustained firing rates (*Chan et al., 2005*; *Surmeier et al., 2005*; *Mastro et al., 2014*; *Abdi et al., 2015*; *Deister et al., 2013*), are well positioned to influence $E_{GABA}$ in the SNr and hence to impact SNr processing of GABAergic inputs from other sources. In particular, our results predict that changes in baseline GPe output will modulate the synchrony between SNr neurons coupled through local GABAergic collaterals and can induce or suppress low frequency oscillations in SNr firing. We present data from experiments involving optogenetic stimulation of GPe terminals in SNr supporting this prediction. Moreover, we find that GPe outputs should be able to tune the effectiveness of GABAergic inputs to the SNr from the Str, which may impact the timing of decisions released by pauses in SNr firing.

From a naive perspective, the excitatory and biphasic inhibitory-to-excitatory SNr responses that we observed following stimulation of GPe and Str projections are surprising, since GABAergic synapses are typically considered as inhibitory and the slice preparation used in our experiments largely

eliminates the possibility of disinhibitory network effects. Excitatory and biphasic GABAergic effects are not unprecedented, however, as they have been reported in other brain regions (*Haam et al., 2012*; *Astorga et al., 2015*). Furthermore, from a theoretical perspective, these GABAergic responses are relatively well understood (see *Dayan and Abbott, 2001*; *Doyon et al., 2011* for reviews). The direction (inhibitory versus excitatory) of the GABAergic current ($I_{GABA}$) depends on the value of $E_{GABA}$ relative to the membrane potential ($V_m$) when GABAA receptors are activated. As such, excitatory responses are expected to result from a GABAergic reversal potential ($E_{GABA}$) that is depolarized close to or above the action potential threshold of a given neuron, while biphasic inhibitory-to-excitatory responses are expected to be mediated by a relatively rapid $Cl^-$ accumulation and ongoing depolarization of $E_{GABA}$ during the arrival of GABAergic inputs, which may be accelerated in small dendritic compartments. In keeping with this idea, stimulation of striatal inputs to SNr in mouse brain slices at a slower rate of 2 Hz yielded consistent initial inhibitory effects rather than the diversity of SNr responses we observed (*Simmons et al., 2018*). It is also possible that sustained stimulation of GPe and Str terminals may yield slow short-term depression that contributes to gradual changes in SNr firing rates, but this would not explain the biphasic SNr responses. Similarly, inhibition could recruit additional currents that are activated by hyperpolarization, such as low voltage-activated $Ca^{2+}$, persistent sodium, or hyperpolarization-activated cyclic nucleotide-gated (HCN) channels, for example. A subset of these currents could theoretically combine to explain the biphasic but not the purely excitatory responses. The data in *Figure 6* and *Figure 6—figure supplements 1* and *2* show a greater proportion of immediate excitatory SNr responses to GPe stimulation than to Str stimulation. This observation suggests that baseline $E_{GABA}$ may be more depolarized at the soma than in the dendrites in SNr neurons, perhaps due to the higher spike rate of GPe than of Str, the preferential dendritic localization of the KCC2 pump in SNr neurons (*Gulácsi et al., 2003*), the basket-like nature of GPe synapses on the SNr soma (*Smith and Bolam, 1991*), or other factors.

As one possible implication of depolarization of $E_{GABA}$, experiments in rodent epilepsy models have revealed that seizure-like events are preceded by surges in interneuron activity that depolarize $E_{GABA}$, sparking a positive feedback loop that can result in runaway activity (*Lillis et al., 2012*; *Kaila et al., 2014*). Interestingly, $E_{GABA}$ has been found to exhibit a strong sensitivity to changes in factors that can affect $Cl^-$ levels (*Kaila et al., 2014*) some of which, such as KCC2-mediated $Cl^-$ extrusion (*Sivakumaran et al., 2015*; *Moore et al., 2017*; *Schulte et al., 2018*; *Titz et al., 2015*), may be tunable by cellular signaling pathways (*Titz et al., 2015*). According to our model, compromised KCC2 function would likely depolarize $E_{GABA}$, slowing or even preventing decision-making. More generally, our results support the idea that GPe output itself could be modulated to tune SNr processing, related to decision speeds or other functions, in condition-specific ways (see *Figure 14*).

Our experiments characterizing SNr responses to optogentic stimulation of GPe and Str GABAergic projections were done in in vitro slice preparations. The literature includes conflicting ideas about whether $E_{GABA}$ is depolarized or hyperpolarized in vitro relative to in vivo conditions. Relatively hyperpolarized $E_{GABA}$ may arise in slice preparations due to severed synaptic projections, which result in an overall reduction of synaptic transmission and, consequently, reduced tonic chloride conductance and load (*Doyon et al., 2011*). Alternatively, $E_{GABA}$ may be depolarized in vitro because tissue damage may compromise KCC2 pump function and other control mechanisms (*Nabekura et al., 2002*; *Herbison and Moenter, 2011*). Indeed, the diversity in responses to inputs across SNr neurons (*Figure 6*) may relate to differences in slicing-induced damage and corresponding baseline $E_{GABA}$ values. Because spiking in the SNr is asynchronous in control animals (*Deransart et al., 2003*; *Willard et al., 2019*), our model would predict that $E_{GABA}$ should be close to —55 mV in vivo (*Figure 8A4*). This value may be depolarized relative to values occurring in vitro, where $E_{GABA}$ has been measured at values in the range from –75 mV to –55 mV (*Giorgi et al., 2007*; *Connelly et al., 2010*; *Higgs and Wilson, 2016*; *Simmons et al., 2018*). If $E_{GABA}$ is depolarized in vivo, then we would also expect to see an increase in the number of SNr neurons that have excitatory responses to optogenetic stimulation of GABAergic projections from GPe neurons of the indirect pathway and Str projections from the direct pathway, relative to our results in vitro (*Figure 6*). Consistent with this prediction, previous in vivo experiments (*Freeze et al., 2013*) found that optogenetic stimulation of D1 Str neurons resulted in excitatory responses in 55% (15 of 27) of SNr neurons. A final consideration relating to our slice experiments is that we did not block excitatory or cholinergic inputs. Thus, related network effects theoretically could have contributed to the SNr responses, although there are no known sources for such effects in the slices that we studied.

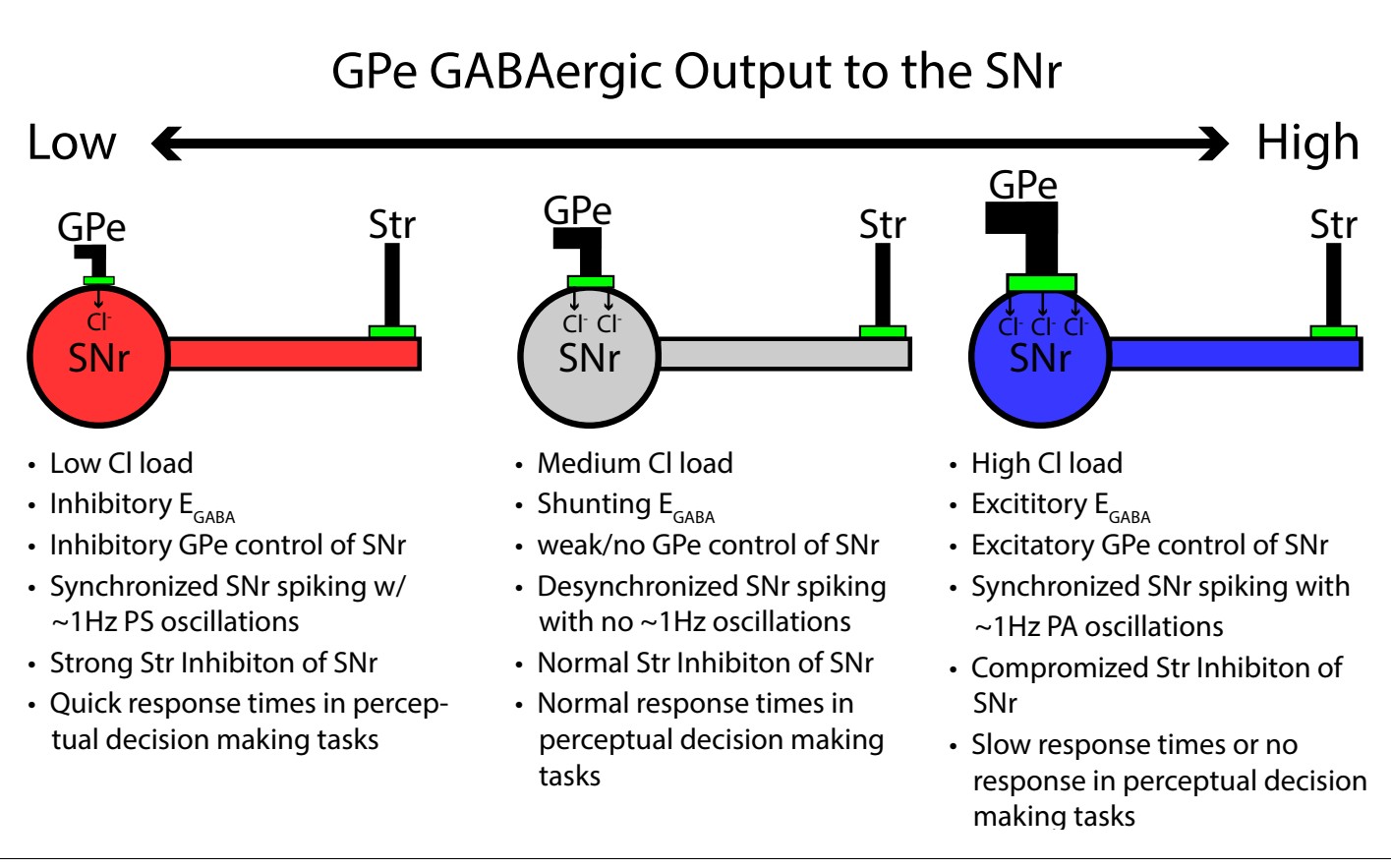

**Figure 14.** Summary figure/cartoon - GPe output provides tonic Cl load tuning SNr synchrony and the strength of Str inhibition.

The impact of GABAergic inputs from GPe on synchrony within SNr predicted by our model is consistent with a previous study that examined the effect of $E_{GABA}$ on dynamics of a bidirectionally coupled neuron pair (*Jeong and Gutkin, 2007*). The previous work also exploited PRCs for its analysis but was done using simpler models, in the context of weak coupling, and did not consider the unidirectional case. In fact, given the sparsity of synaptic collaterals within SNr (*Higgs and Wilson, 2016*; *Simmons et al., 2018*), we expect that unidirectional connectivity between SNr neurons would be the dominant motif observed. Thus, our model suggests that GPe firing rates could tune the level of synchrony in SNr, with oscillations emerging when $E_{GABA}$ is below the afterhyperpolarization potential.

The oscillations that we predict will arise in SNr neurons are slower than the β oscillations often discussed in the context of parkinsonism. These slow oscillations are consistent with previous results in anesthetized animals (*Walters et al., 2007*) and arise in recently reported experiments (*Whalen et al., 2020*) and in the data presented here. Our results, based on the amplitude and shape of PRCs, predict that oscillation frequency will vary with $E_{GABA}$ and with the strength of synapses between SNr neurons (*Figure 9*) but never reach frequencies in the β band. Various data, simulations and theory suggest different changes in PRC shape with neuronal firing rate (*Tsubo et al., 2007*; *Phoka et al., 2010*; *Couto et al., 2015*; *Ermentrout and Terman, 2010*). Simulations of our model SNr neuron showed a reduction in PRC amplitude with increased firing rate, up to saturation around 25 Hz, which would lead to the need for more synaptic inputs to occur to achieve one full passage along the PRC (e.g., *Figure 9*). This explains why at higher firing rates, although more synaptic inputs occur in a given time, the slow oscillation frequency does not significantly increase; see *Figure 10—figure supplement 1*. The mechanism underlying the changes in our model neuron's PRC with firing rate likely depends on the particular currents included but remains for future investigation.

The precise functions of Str inputs to SNr neurons remain unknown. Although there is significant literature supporting a role for these inputs in action selection or initiation, there are certainly other possibilities. One such idea is that Str inputs encode movement velocity and the resulting SNr firing rate encodes spatial position (*Kim et al., 2014*; *Bartholomew et al., 2016*; *Barter et al., 2015*). If we apply our modeling results to this view, then we predict that the $Cl^-$ load from the GPe, by tuning SNr responses to GABAergic inputs from Str, could impact velocity with which selected movements are performed. On the other hand, we do not expect that Str inputs would tune $E_{GABA}$ in SNr and SNr synchrony, as we predict for GPe inputs. This difference arises due to the lower Str baseline firing rate, which would have less impact on $Cl^-$ load, and the dendritic targeting of Str inputs to SNr, which would not induce a strong effect at the soma. We note that these Str neuron baseline firing rates are significantly lower than the stimulation frequency in our optogenetic activation of Str terminals. Furthermore, in contrast to the simplifications in our simulations, Str inputs to SNr are distributed over an extended, branched dendritic tree, such that individual branches may receive only very low rate inputs in vivo. Nonetheless, similar $Cl^-$ dynamics and SNr responses could result from a collection of lower-rate Str inputs in natural settings, albeit with heterogeneity over specific levels of $Cl^-$ across different dendritic branches.

In mice, DA depletion increases SNr synchrony (*Willard et al., 2019*) and promotes slow oscillations (*Whalen et al., 2020*). In our model, this may be explained by a hyperpolarizing shift in $E_{GABA}$ under these conditions, presumably driven by a reduction in GPe firing rate (*Filion and Tremblay, 1991*; *Boraud et al., 1998*; *Wichmann et al., 2002*) and/or decreased GABAergic synaptic output to the SNr. Therefore, we would predict GABAergic inhibiton to be stronger under DA depletion. This is consistent with previous a previous study which shows that GABAergic inhibition in the SNr is attenuated by activation of D2 receptors (*Martin and Waszczak, 1996*). Additionally, our model predicts that strengthened GABAergic inhibition could enhance the capability of inputs from the Str to pause SNr firing, potentially facilitating action selection. Consistent with this idea, DA depletion has been shown to accelerate saccadic perceptual decisions in humans (*van Stockum et al., 2011*; *van Stockum et al., 2013*).

While our model allows for the simulation of multiple sources of GABA to SNr neurons along with somato-dendritic interactions, short-term synaptic plasticity, and time courses of $[Cl^-]$ and $E_{GABA}$ dynamics, it does omit a variety of additional factors that could impact our predictions. Most significantly, to focus on GABAergic effecs, we ignored STN inputs to SNr neurons. In baseline conditions of ongoing high frequency STN activity, these inputs would help tune SNr excitability but we do not expect them to be relevant for adjusting $Cl^-$ load and $E_{GABA}$; the effects of more patterned STN activity under DA depletion remain to be explored. Secondly, our description of the location of GPe projections on SNr neurons involves some simplification. GPe projections primarily form synapses around the soma but also form synapses on proximal dendrites (*Smith and Bolam, 1991*; *von Krosigk et al., 1992*), which we have ignored. The study conducted by *Smith and Bolam, 1991* found that SNr-projecting GPe neurons formed synapses with the soma and the distal dendrites of 54% and 32% of SNr neurons, respectively. Although our model does not distinguish among the diverse subpopulations of GPe neurons that have been identified (*Mastro et al., 2014*; *Hernández et al., 2015*; *Abdi et al., 2015*), an intriguing possibility for future study is that different subsets of GPe neurons may project to different sites on SNr neurons, allowing for separable control over local SNr interactions and synchrony versus responses to Str inputs. Along similar lines, we assumed that GABAergic SNr collaterals form somatic as opposed to dendritic synapses. We also did not model non-neuronal cells such as glia that can affect extracellular ion concentrations, which could reduce the amplitude of the effects that we describe; the variability in extracellular concentrations of ions other than $Cl^-$ such as $K^+$, which could affect SNr excitability; slower components of synaptic depression that, if present, may yield a gradual weakening of inhibition over several seconds; and direct effects of DA and other neuromodulators.

We have cited and shown that our results are consistent with a range of experimental data. To really pin down the relevance of the proposed mechanisms, future experiments would need to be performed to measure intracellular $[Cl^-]$ or $E_{GABA}$ itself. For the latter, it may be possible to perform perforated patch recordings and measure $E_{GABA}$ as a function of GPe firing rate, but these experiments are challenging and may not be possible in dendrites. A more attainable first step would be to repeat the in vitro stimulation experiments under pharmacological blockade of KCC2, to check for a resulting bias toward excitatory and biphasic responses, and in the presence of KCC2

enhancers, to check for a shift toward inhibitory responses (*Hamidi and Avoli, 2015*). Another option would be to break into whole cell mode and repeat stimulation with control of $E_{Cl}$ using the chloride from the pipette, to check if excitatory and biphasic effects can be eliminated. A final option is to express halorhodopsin in the SNr, and directly control local chloride flow into the cell. If they are borne out by future experiments, the findings of this study may have implications outside of the SNr, as GABA$_A$ is a major neurotransmitter in the CNS.

# Materials and methods

## Model description

Model SNr neurons were developed that each feature both a somatic and a dendritic compartment and incorporate Hodgkin-Huxley style conductances adapted from previously described models and/or experimental data (*Xia et al., 1998*; *Zhou et al., 2008*; *Corbit et al., 2016*; *Doyon et al., 2015*). The membrane potentials for the somatic ($V_S$) and dendritic compartments ($V_D$) are given by the following differential equations:

$$C_S \frac{dV_S}{dt} = -I_{Na} - I_{NaP} - I_K - I_{Ca} - I_{SK} - I_{Leak} - I_{GABA}^S - I_{DS} + I_{APP} \tag{1}$$

$$C_D \frac{dV_D}{dt} = -I_{TRPC3} - I_{GABA}^D - I_{SD} \tag{2}$$

where $C_S = 100 pF$ and $C_D = 40 pF$ are the capacitances for the somatic and dendritic compartments. The currents in each compartment are represented by $I_i$ where $i$ denotes the current type. The somatic compartment features the essential spike generating currents as well as several others: fast Na$^+$ current ($I_{Na}$), persistent Na$^+$ current ($I_{NaP}$), delayed rectifying K$^+$ current ($I_K$), Ca$^{2+}$ current ($I_{Ca}$), Ca$^{2+}$-activated K$^+$ current ($I_{SK}$), and leak current ($I_{Leak}$) as well as a synaptic current which represents the GABAergic input from the GPe neurons of the indirect pathway ($I_{GABA}^S$). $I_{APP}$ denotes an applied current injected from an electrode. The dendritic compartment contains a current from a transient receptor potential channel 3 (TRPC3) ($I_{TRPC3}$) and a synaptic current ($I_{GABA}^D$), which represents the GABAergic input from the striatal neurons of the direct pathway. $I_{TRPC3}$ contributes to depolarization of the SNr neuron (*Zhou et al., 2008*) and was included in anticipation of future work to consider the dopamine depleted regime, in which this channel may be altered (*Zhou et al., 2009*). The two additional currents $I_{DS}$ and $I_{SD}$ are coupling terms that represent the current from the dendrite into the soma and from the soma into the dendrite, respectively. The currents are defined as follows:

$$I_{Na} = g_{Na} \cdot m_{Na}^3 \cdot h_{Na} \cdot s_{Na} \cdot (V_S - E_{Na}) \tag{3}$$

$$I_{NaP} = g_{NaP} \cdot m_{NaP}^3 \cdot h_{NaP} \cdot (V_S - E_{Na}) \tag{4}$$

$$I_K = g_K \cdot m_K^4 \cdot h_K \cdot (V_S - E_K) \tag{5}$$

$$I_{Ca} = g_{Ca} \cdot m_{Ca} \cdot h_{Ca} \cdot (V_S - E_{Ca}) \tag{6}$$

$$I_{SK} = g_{SK} \cdot m_{SK} \cdot (V_S - E_K) \tag{7}$$

$$I_{Leak} = g_{Leak} \cdot (V_S - E_{Leak}) \tag{8}$$

$$I_{GABA}^S = g_{GABA}^S \cdot (V_S - E_{GABA}^S) \tag{9}$$

$$I_{DS} = \frac{g_C}{\alpha_C} \cdot (V_S - V_D) \tag{10}$$

$$I_{TRPC3} = g_{TRPC3} \cdot (V_D - E_{TRPC3}) \tag{11}$$

$$I_{GABA}^D = g_{GABA}^D \cdot (V_D - E_{GABA}^D) \tag{12}$$

$$I_{SD} = \frac{g_C}{1 - \alpha_C} \cdot (V_D - V_S), \tag{13}$$

where $g_i$ is the maximum conductance, $E_i$ is the reversal potential, and $m_i$ and $h_i$ are gating variables for channel activation and inactivation for each current $I_i$. $s_{Na}$ is an additional inactivation term governing spike-frequency adaptation. The parameter $\alpha_C = 0.714$ is the ratio of somatic and total capacitances. The GABAergic synaptic conductances $g_{GABA}^S$, $g_{GABA}^D$ are variable and will be defined below. The values used for the $g_i$ and $E_i$ are given in **Table 1**.

Activation ($m_i$) and inactivation ($h_i$, $s_i$) of voltage-dependent channels are described as follows:

$$\frac{dz_i}{dt} = \frac{z_i^\infty - z_i}{\tau_{z_i}}, \quad i = \{Na, NaP, K, Ca\}, \quad z = \{m, h, s\}. \tag{14}$$

Steady-state (in)activation functions and their time constants ($\tau_{z_i}$) are described by:

$$z_i^\infty(V) = \frac{1}{1 + e^{-(V - z_{1/2}^i)/k_{z_i}}}, \tag{15}$$

$$\tau_{z_i}(V) = \tau_{z_i}^0 + \frac{\tau_{z_i}^1 - \tau_{z_i}^0}{e^{(\tau_{1/2}^i - V)/\sigma_{z_i}^0} + e^{(\tau_{1/2}^i - V)/\sigma_{z_i}^1}}. \tag{16}$$

The parameters for these currents are given in **Table 1** and were adapted from **Corbit et al., 2016**.

Activation of the small conductance calcium-activated potassium channels (SK) is instantaneous and depends on the intracellular calcium concentration ($[Ca]_i$):

$$m_{SK}([Ca]_{in}) = \left(1 + \left(\frac{k_{SK}}{[Ca]_{in}}\right)^{n_{SK}}\right)^{-1}, \tag{17}$$

where $k_{SK}$ represents the half-activation $Ca^{2+}$ concentration and $n_{SK}$ is the Hill coefficient. The parameters are given in **Table 1** and were taken from **Xia et al., 1998**.

The intracellular calcium concentration is determined by the balance of $Ca^{2+}$ influx carried by $I_{Ca}$ and efflux via the $Ca^{2+}$ pump. In the model, $I_{Ca}$ and $I_{SK}$ are only expressed in the soma and therefore $[Ca]_{in}$ dynamics is only simulated in the somatic compartment. The dynamics of $[Ca]_{in}$ is described by the following equation:

$$\frac{d[Ca]_{in}}{dt} = -\alpha_{ca} \cdot I_{Ca} - ([Ca]_{in} - Ca_{min})/\tau_{Ca}, \tag{18}$$

where $\alpha_{ca} = 1.0 \cdot 10^{-8} \, mM/fC$ is a conversion factor relating current and rate of change in $[Ca]_{in}$, $\tau_{Ca} = 250 \, ms$ is the time constant for the $Ca^{2+}$ extrusion and $Ca_{min} = 5.0 \cdot 10^{-8} \, mM$ is the minimum calcium concentration, where the $Ca^{2+}$ pump turns off. Because of the balance between $Ca^{2+}$ efflux from the pump and influx from $I_{Ca}$ activation, these parameters result in a typical value for $[Ca]_{in}$ of about $2.5 \cdot 10^{-4} \, mM$ in our simulations.

## Synaptic dynamics

The GABAergic synaptic conductance in the somatic ($g_{GABA}^S$) and dendritic $g_{GABA}^S$ compartments are described by the following equations:

$$\frac{dg_{GABA}^S}{dt} = -\frac{g_{GABA}^S}{\tau_{GABA}^S} + W_{GABA}^{GPe} \cdot D \cdot \delta(t - t_n) + W_{GABA}^{SNr} \cdot \delta(t - t_m), \tag{19}$$

and

$$\frac{dg^D_{GABA}}{dt} = -\frac{g^D_{GABA}}{\tau^D_{GABA}} + W^{Str}_{GABA} \cdot F \cdot \delta(t - t_l),$$ (20)

where $\tau^{\{S,D\}}_{GABA}$ is the exponential decay time constant for the somatic and dendritic compartments,

**Table 1.** Ionic channel parameters.

| Channel | Parameters | | |
|---|---|---|---|
| $I_{Na}$ | $g_{Na} = 35\,nS/pF$ | $E_{Na} = 50.0\,mV$ | |
| | $m_{1/2} = -30.2\,mV$ | $k_m = 6.2\,mV$ | |
| | $\tau^0_m = 0.05\,ms$ | $\tau^1_m = 0.05\,ms$ | $\tau^m_{1/2} = 1\,mV$ |
| | $\sigma^0_m = 1\,mV$ | $\sigma^1_m = 1\,mV$ | |
| | $h_{1/2} = -63.3\,mV$ | $k_h = -8.1\,mV$ | |
| | $\tau^0_h = 0.59\,ms$ | $\tau^1_h = 35.1\,ms$ | $\tau^h_{1/2} = -43.0\,mV$ |
| | $\sigma^0_h = 10\,mV$ | $\sigma^1_h = -5\,mV$ | |
| | $s_{1/2} = -30.0\,mV$ | $k_s = -0.4\,mV$ | |
| | $\tau^0_s = 10\,ms$ | $\tau^1_s = 50\,ms$ | $\tau^s_{1/2} = -40\,mV$ |
| | $\sigma^0_s = 18.3\,mV$ | $\sigma^1_s = -10\,mV$ | $s_{min} = 0.15$ |
| $I_{NaP}$ | $g_{NaP} = 0.175\,nS/pF$ | | |
| | $m_{1/2} = -50.0\,mV$ | $k_m = 3.0\,mV$ | |
| | $\tau^0_m = 0.03\,ms$ | $\tau^1_m = 0.146\,ms$ | $\tau^m_{1/2} = -42.6\,mV$ |
| | $\sigma^0_m = 14.4\,mV$ | $\sigma^1_m = -14.4\,mV$ | $m_{min} = 0.0$ |
| | $h_{1/2} = -57.0\,mV$ | $k_h = -4.0\,mV$ | |
| | $\tau^0_h = 10.0\,ms$ | $\tau^1_h = 17.0\,ms$ | $\tau^h_{1/2} = -34.0\,mV$ |
| | $\sigma^0_h = 26.0\,mV$ | $\sigma^1_h = -31.9\,mV$ | $h_{min} = 0.154$ |
| $I_K$ | $g_K = 50\,nS/pF$ | $E_K = -90.0\,mV$ | |
| | $m_{1/2} = -26\,mV$ | $k_m = 7.8\,mV$ | |
| | $\tau^0_m = 0.1\,ms$ | $\tau^1_m = 14.0\,ms$ | $\tau^m_{1/2} = -26.0\,mV$ |
| | $\sigma^0_m = 13.0\,mV$ | $\sigma^1_m = -12.0\,mV$ | |
| | $h_{1/2} = -20.0\,mV$ | $h_m = -10.0\,mV$ | |
| | $\tau^0_h = 5.0\,ms$ | $\tau^1_h = 20.0\,ms$ | $\tau^h_{1/2} = 0.0\,mV$ |
| | $\sigma^0_h = 10.0\,mV$ | $\sigma^1_h = -10.0\,mV$ | $h_{min} = 0.6$ |
| $I_{Ca}$ | $g_{Ca} = 0.7\,nS/pF$ | $E_{Ca} = 13.27 \cdot ln(Ca_{out}/Ca_{in})$ | |
| | $Ca_{out} = 4.0\,mM$ | $Ca_{in}$, see **Equation 18** | |
| | $m_{1/2} = -27.5\,mV$ | $k_m = 3.0\,mV$ | $\tau_m = 0.5\,ms$ |
| | $h_{1/2} = -52.5\,mV$ | $k_h = -5.2\,mV$ | $\tau_h = 18.0\,ms$ |
| $I_{SK}$ | $k_{SK} = 0.4\,mM$ | $n_{SK} = 4$ | $\tau_{sk} = 0.1\,mS$ |
| $I_{Leak}$ | $g_{Leak} = 0.04\,nS/pF$ | $E_{Leak} = -60\,mV$ | |
| $I^S_{GABA}$ | $W^{GPe}_{GABA} = 0.2\,nS/pF$ | $E^S_{GABA}$, see **Equation 23** | $\tau_{SynE} = 3.0\,ms$ |
| | $D_0 = 1.0$ | $\alpha_D = 0.565$ | $\tau_D = 1000\,ms$ |
| | $D_{min} = 0.67$ | $W^{SNr}_{GABA} = 0.1\,nS/pF$ | |
| $I_{SD}, I_{DS}$ | $g_C = 26.5\,nS$ | | |
| $I_{TRPC3}$ | $g_{TRPC3} = 0.1\,nS/pF$ | $E_{TRPC3} = -37.0\,mV$ | |
| $I^D_{GABA}$ | $W^{Str}_{GABA} = 0.4\,nS/pF$ | $E^D_{GABA}$, see **Equation 23** | $\tau^D_{GABA} = 7.2\,ms$ |
| | $F_0 = 0.145$ | $\alpha_F = 0.125$ | $\tau_F = 1000\,ms$ |

$W_{GABA}^{\{GPe,SNr,Str\}}$ is the synaptic weight of inputs from the GPe, SNr, and Str. $\delta(.)$ represents the Kronecker delta function, $t$ is time, and $t_{\{n,m,l\}}$ represent the times that inputs $n, m, l$ are received from GPe, SNr, and Str, respectively. The functions $D$ and $F$ are scaling factors representing short-term synaptic depression and facilitation, which were simulated using an established mean-field model of short-term synaptic depression/facilitation (*Abbott, 1997*; *Dayan and Abbott, 2001*; *Morrison et al., 2008*) as follows:

$$\frac{dD}{dt} = \frac{D_0 - D}{\tau_D} - \alpha_D (D - D_{min}) \cdot \delta(t - t_i), \tag{21}$$

and

$$\frac{dF}{dt} = \frac{F_0 - F}{\tau_F} + \alpha_F (1 - F) \cdot \delta(t - t_k). \tag{22}$$

The parameters for $D_0$, $\tau_D$, $\alpha_D$, $D_{min}$, $F_0$, $\tau_F$, and $\alpha_F$ are listed in *Table 1* and were chosen to empirically match experimental data from *Connelly et al., 2010*, see *Figure 2*.

## Chloride and $E_{GABA}$ dynamics

GABA$_A$ receptors are permeable to both $Cl^-$ and $HCO3^-$ ions. Therefore, the reversal potential $E_{GABA}$ is a function of ion concentration gradients for both of these substances and is determined by the Goldman-Hodgkin-Katz voltage equation:

$$E_{GABA} = \frac{RT}{F} \cdot ln\left(\frac{4[Cl^-]_{in} + [HCO_3^-]_{in}}{4[Cl^-]_{out} + [HCO_3^-]_{out}}\right), \tag{23}$$

where $R = 8.314\,J/(molK)$ is the universal gas constant; $T = 308\,K$ is temperature; $F = 96.485\,kC/mol$ is the Faraday constant. The concentrations $[Cl^-]_{out} = 120\,mM$, $[HCO_3^-]_{in} = 11.8\,mM$, $[HCO_3^-]_{out} = 25.0\,mM$ are fixed parameters representing the extracellular $Cl^-$ and intracellular and extracellular $HCO3^-$ concentrations, respectively. Parameters were adapted from *Doyon et al., 2015*. The intracellular $Cl^-$ concentration in the somatic ($[Cl^-]_{in}^S$) and dendritic ($[Cl^-]_{in}^D$) compartments is dynamic and is determined by the balance of $Cl^-$ influx through GABAergic synapses ($I_{GABA}$) and efflux via the KCC2 $Cl^-$ extruder. In both compartments, the dynamics of $[Cl^-]_{in}$ is governed by the following equation:

$$\frac{d[Cl^-]_{in}}{dt} = -\alpha_{Cl} \cdot [g_{KCC2} \cdot (E_{Cl} - E_k) - \chi \cdot (g_{GABA} + g_{GABA}^{Tonic}) \cdot (V - E_{Cl})], \tag{24}$$

$$\chi = \frac{E_{HCO3} - E_{GABA}}{E_{HCO3} - E_{Cl}}, \quad E_{Cl} = \frac{RT}{F} \cdot ln\left(\frac{[Cl^-]_{in}}{[Cl^-]_{out}}\right), \quad \text{and} \quad E_{HCO3} = \frac{RT}{F} \cdot ln\left(\frac{[HCO3^-]_{in}}{[HCO3^-]_{out}}\right). \tag{25}$$

In the previous equations, $\alpha_{Cl}$ is a conversion factor relating current and rate of change in $[Cl^-]_{in}$, $g_{KCC2}$, $g_{GABA}$ and $g_{GABA}^{Tonic}$ are the conductance of the KCC2 $Cl^-$ extruder, the GABAergic conductance, and the tonic chloride load. $\chi$ describes the fraction of the $GABA_A$ current that is carried by $Cl^-$ ions, and $V$ represents the membrane potential of the specific compartment. The dynamics of $Cl^-$ are simulated separately for the somatic ($[Cl^-]_{in}^S$) and dendritic ($[Cl^-]_{in}^D$) compartments, which have distinct $\alpha_{Cl}$ values. Under the assumptions that neuronal capacitance scales with surface area as $0.89\,\mu F/cm^2$(*Gentet et al., 2000*), that the nuclear-cytoplasmic volume ratio of the SNr soma is 1:1 (*Paloff et al., 1989*), and that the somatic capacitance is $100\,pF$, we obtain $\alpha_{Cl}^{soma} = 1.77 \cdot 10^{-7}\,mM/fC$. Similarly, assuming that the dendrite and soma have the same membrane thickness and electrical permittivity (which set the scaling of capacitance to surface area), that the dendritic capacitance is $40\,pF$, that the dendrite is a cylinder of radius $2\,\mu m$, and that the full dendritic volume is accessible to ions, we obtain $\alpha_{Cl}^{dend} = 2.2125 \cdot 10^{-7}\,mM/fC$. In both compartments $g_{KCC2}$ and $g_{GABA}^{Tonic}$ are parameters which are varied to tune $E_{GABA}$. Specifically, $g_{KCC2}$ is varied from 0.0 to $0.4\,nS/pF$ and $g_{GABA}^{Tonic}$ is from 0.0 to $1.0\,nS/pF$. $E_K$ is fixed and can be found in *Table 1*. This mathematical description of $Cl^-$ dynamics was adapted from *Doyon et al., 2015*.

## Phase response curves

The data for calculating the phase response curves were generated by simulating transient GABAergic inputs to the somatic compartment every $2\,s$ plus a randomly generated variation of 0 to $100\,ms$. The dataset was post-processed in Matlab and for each simulated GABAergic input, the change in phase relative to the input phase was extracted. Equations for the PRCs were generated using a fourth order polynomial fit.

Bidirectional network: Phase on the horizontal axis is defined in a frame relative to the phase of neuron 1. In other words, to compute the PRC of neuron 2, we consider the effect of an input from neuron 1 to neuron 2 when neuron 2 is at different phases; the fact that neuron one is supplying the input means that the phase of neuron 1 is 1. To compute the PRC of neuron 1, we should still think of the phase of neuron 1 as being 1 (or equivalently 0), but now neuron two is the neuron providing the input. As a result, the PRC for neuron 1 ends up being given by reflecting the PRC for neuron 2 about 0.5.

For example, suppose that the phase of neuron 2 is altered by an amount $\Delta\phi$ if it receives an input when it is at phase 0.8, such that the PRC of neuron 2 takes the value $\Delta\phi$ at phase $\phi = 0.8$. Note that at $\phi = 0.8$, neuron 2 lags neuron 1 by a phase of 0.2. Now, at what phase should the PRC for neuron 1 take the value $\Delta\phi$? To answer this question, we must determine the phase of neuron 2 when it spikes, given that neuron 1 lags neuron 2 by 0.2. But since the phase of neuron 1 is 0, we simply conclude that the value $\Delta\phi$ occurs on the PRC of neuron 1 at $\phi = 0.2$ (i.e., at $\phi = 1 - 0.8$); see *Figure 8—figure supplement 1*.

## SNr network construction

As mentioned above, the SNr is a sparsely connected network where each neuron is estimated to receive between 1–4 inputs from neighboring SNr neurons (*Higgs and Wilson, 2016*). To represent sparse connectivity in our simulated 100 neuron SNr network (see *Figure 13A*), *Equation (19)* for $g_{GABA}^{S}$ was slightly modified such that the somatic GABAergic conductance in the $i^{th}$ neuron in the population is described by the following equation:

$$g_{GABA}^{S} = \sum_{j \neq i} \sum_{n} W_{j,i}^{SNr} \cdot C_{ji} \cdot H(t - t_{j,n}) \cdot e^{-(t - t_{j,n})/\tau_{GABA}^{S}}, \qquad (26)$$

where $W_{j,i}^{SNr}$ is the weights of the SNr to SNr synaptic connection from source neuron *j* to the target neuron *i*. $C_{ji}$ is a connectivity matrix where $C_{ji} = 1$ if neuron j makes a synapse on neuron i, and $C_{ji} = 0$ otherwise. $H(.)$ is the Heaviside step function, and $t$ denotes time. $t_{j,n}$ is the time at which the $n^{th}$ action potential is generated in neuron *j* and reaches neuron *i*. Sparse connectivity in the model was achieved by randomly assigning the vales of $C_{ji}$ such that the probability of any connection between neuron *i* and *j* being one is equal to 0.02. Heterogeneity in the network was introduced by uniformly distributing the weights of SNr connections such that $W_{j,i}^{SNr} = U(0, 0.1)\,nS/pF$. Additionally, in order to match in vivo data (*Freeze et al., 2013*; *Mastro et al., 2017*; *Willard et al., 2019*) the baseline firing rate was increased to $\approx 25\,Hz$ by setting $g_{Glut}^{D} = U(0.02, 0.12)\,nS$.

Additionally, diffusion of $Cl^{-}$ between the somatic and dendritic compartments is incorporated into the network model. This was simulated by the addition of the exponential decay terms $-([Cl^{-}]_{in}^{S} - [Cl^{-}]_{in}^{D})/(\tau_{SD})$ and $-([Cl^{-}]_{in}^{S} - [Cl^{-}]_{in}^{S})/(\tau_{DS})$ into *Equation (24)* for the somatic and dendritic compartments respectively. The parameters $\tau_{SD} = 200\,ms$ and $\tau_{DS} = 80\,ms$ are exponential decay time constants. These values reflect a higher chloride load in the soma than the dendrite due to tonic GPe inputs to the soma as well as the preferential expression of KCC2 in dendrites of SNr neurons (*Gulácsi et al., 2003*). Because of this configuration, it is likely that somatic $Cl^{-}$ will diffuse from the soma to the dendrite, ultimately affecting dendritic $E_{GABA}$. The specific time constants, $\tau_{SD}$ and $\tau_{DS}$, were set at values for which $E_{GABA}$ in the dendrite was hyperpolarized relative to the soma by approximately $2.5 - 8\,mV$, as reported in the literature (*Connelly et al., 2010*; *Lavian and Korngreen, 2016*) and are not intended to reflect or match rates of axial $Cl^{-}$ diffusion.

## Data analysis and definitions

Data generated from simulations was post-processed in Matlab (Mathworks, Inc). An action potential was defined to have occurred in a neuron when its membrane potential $V_m$ increased through

$-35 mV$. For characterization of the paired pule ratios of simulated GPe and Str inputs (**Figures 2** and **3**), the IPSC/IPSP amplitude is defined as the absolute value of the difference between current/ potential immediately before the start of the synaptic input and the local maximum occurring in a 10 ms window following the synaptic input. Histograms of population activity were calculated as the number of action potentials per 20 ms bin per neuron with units of $APs/(s \cdot neuron)$.

The response of SNr neurons to optogenetic stimulation of GPe and Str terminals were categorized by breaking up the full 10 s stimulation period into bins. The first $1 s$ was broken up into 1/3 s bins. The rest of the period was broken into 1 s bins. The spiking in each bin was then compared to baseline using a Mann-Whitney U test with Bonferroni correction where $p<0.00416$ was considered statistically significant. Each response category was defined as follows: (1) Complete Inhibition: at most five spikes in the full 10 s period, (2) Partial Inhibition: at least one bin is statistically less than baseline and no bins are excited, (3) No Effect: no bins are statistically different than baseline, (4) Excitation: at least one bin is statistically above baseline and no bins are less than baseline, (5) Biphasic: at least one bin is statistically below and one above baseline. In order to identify pauses that are longer than can be accounted for by short-term synaptic dynamics, the 'long pause' was defined as any pause in spiking that continues after 10 stimulus pulses (steady state is reached after roughly five pulses), which equates to 1000 ms, 500 ms, 250 ms and 125 ms for stimulation at 10 Hz, 20 Hz, 40 Hz and 60 Hz, respectively.

### Integration methods

All simulations were performed locally on an 8-core Linux-based operating system. Simulation software was custom written in C++. Numerical integration was performed using the first-order Euler method with a fixed step-size ($\Delta t$) of 0.025 ms. All model codes will be made freely available through the ModelDB sharing site hosted by Yale University upon publication of this work.

### Animals

All experiments were conducted in accordance with guidelines from the National Institutes of Health and with approval from the Carnegie Mellon University Institutional Animal Care and Use Committee. Male and female mice on a C57BL/6J background aged 8–15 weeks were used. Animals were caged in groups of 5 or fewer with food and water always available. Light and dark were alternated in a cycle of 12 hr each.

### Slice electrophysiology

Coronal slices containing SNr (300 μm) were prepared using a VT1000S vibratome (Leica Microsystems) from brains of 6–9 week-old (both male and female) mice that had received ChR2 viral injections 2–4 weeks prior. Slices were cut in carbogenated HEPES ACSF containing the following (in mM): 20 HEPES, 92 NaCl, 1.2 NaHCO₃, 2.5 KCl, 1 MgSO₄, 2 CaCl₂, 30 NaH₂PO₄, 25 glucose, pH 7.25. Slices were allowed to recover for 15 min at 33°C in a chamber filled with N-methyl-D-glucamine-HEPES recovery solution (in mM): 93 N-methyl-D-glucamine, 2.5 KCl, 1.2 NaH₂PO₄, 30 NaHCO₃, 20 HEPES, 25 glucose, 10 MgSO₄, 0.5 CaCl₂. Slices were then held at room temperature for at least 1 hr before recording. Recordings were conducted at 33°C in carbogenated ACSF (in mM) as follows: 125 NaCl, 26 NaHCO₃, 1.25 NaH₂PO₄, 2.5 KCl, 12.5 glucose, 1 MgSO₄, and 2 CaCl₂. Data were collected with a MultiClamp 700B amplifier (Molecular Devices) and ITC-18 analog-to-digital board (HEKA) using Igor Pro software (Wavemetrics, RRID:SCR_000325) and custom acquisition routines (Recording Artist; Richard C. Gerkin, Phoenix). Data were collected at 10 kHz and digitized at 40 kHz. Electrodes were made from borosilicate glass (pipette resistance, 2–6 M). The pipette solution consisted of (in mM): 130 KMeSO₃, 10 NaCl, 2 MgCl₂, 0.16 CaCl₂, 0.5 EGTA, 10 HEPES, 2 Mg-ATP, and 0.3 NaGTP.

### In vivo electrophysiology

Animals were anesthetized with 20 mg/kg ketamine and 6 mg/kg xylazine and placed in a stereotaxic frame (Kopf Instruments). Anesthesia was maintained throughout surgery with 1.0–1.5% isoflurane. All coordinates were measured in mm with AP and ML measured from bregma and DV relative to the dural surface. Injections (200–250 nL) of purified AAV2-DIO-ChR2-EYFP (UNC Vector Core) were performed in the bilateral GPe of Pvalb-2A-Cre transgenic mice (Zeng, Allen Institute). Bregma

coordinates AP: −0.27–0.30 mm, ML: 2.1–2.2 mm, DV: 3.65 mm. To prevent backflow of virus, the pipette was left in the brain for 5 min after completion of the injection. Two to four weeks later, a second surgery was performed to bilaterally deplete dopamine, implant fibers in the GPe for stimulation, implant head bars for recordings, and make bilateral craniotomies over the SNr. For dopamine depletions, holes were drilled over the medial forebrain bundle (MFB, AP: −0.80, ML: ±1.10) and 1 μL of 5 μg/μL 6-OHDA (Sigma-Aldrich) was injected in each side with a GenieTouch Hamilton syringe pump (Kent Scientific). The infusion cannula was left in place for 5 min post-injection before being slowly retracted. Optical fibers for stimulation during recordings were implanted into the bilateral GPe and secured with dental cement in customized plastic holders. For head bar implantation and bilateral craniotomies, the scalp was opened and windows approximately 1.5 × 1.5 mm in size were drilled over SNr (AP: −3.00, ML: ±1.50). A custom-made copper or stainless steel headbar was affixed to the mouse's skull with dental cement (Lang Dental). A well of dental cement was then built around the exposed skull and filled with a silicon elastomer. Upon completion of surgery, animals were injected subcutaneously with 0.5 mg/kg ketofen and placed inside their cage half on/half off a heating pad to recover. Dopamine depleted animals were supplied with trail mix and moistened food to maintain weight and hydration, in addition to their usual food pellets and water bottles, and animals were tracked regularly to ensure proper health and weight.

To perform recordings, mice were head-fixed atop a free-running wheel. After acclimation to head-fixation for ten minutes, the silicon elastomer was removed and craniotomies were cleaned with saline. Using a micromanipulator (Sutter Instruments), a linear microelectrode probe with sixteen channels spaced 50 μm apart (NeuroNexus) was lowered into the SNr craniotomy on one side. After the initial lowering, a ground wire was placed in saline in the dental cement well on the skull. Every time the recording probe was moved, we waited for 10 min before acquiring data to all allowed recordings to stabilize. Spiking (bandpass filtered for 150–8000 Hz, sampled at 40 kHz) and local field potential (bandpass filtered to 0.5–300 Hz, sampled at 1 kHz) recordings were collected through an OmniPlex amplifier (Plexon, Inc) with common median virtual referencing. Simultaneous to these recordings, the mouse's walking speed on the wheel was recorded using an optical mouse and fed to a TTL-pulser which was connected to the OmniPlex amplifier analog input. Optical stimuli were delivered at a power of 1 mW (transmittance through fibers was measured before implanting and confirmed again after fibers were removed postmortem).

Spikes were manually sorted into single units using Offline Sorter (Plexon). For classification as a single unit, the following criteria were set: (1) principal component analysis of waveforms generated a cluster of spikes significantly distinct from other unit or noise clusters (p < 0.05), (2) the J3-statistic was greater than 1, (3) the Davies-Bouldin statistic was less than 0.5, and (4) fewer than 0.15% of ISI's were less than 2 ms. In the case where a unit was lost during recording, it was only used in analysis for the time period when its spike cluster satisfied these criteria, and only if its cluster was present for at least three minutes. Data were then imported into MATLAB (MathWorks) in which all further analysis was performed using custom code except when specified.

After recording, animals were sacrificed and perfused with 4% paraformaldehyde (PFA). The brain was extracted from the skull and stored in PFA for 24 hr then moved to a 30% sucrose solution for at least 24 additional hours. Tissue was sectioned using a freezing microtome (Microm HM 430; Thermo Scientific) and primary antibody incubations were performed on these sections at room temperature for 24 hr. A tyrosine-hydroxylase (TH) antibody (rabbit anti-TH, 1:1000; Pel-Freez) was used to confirm successful dopamine depletion in 6-OHDA-depleted animals. An Iba1 antibody (rabbit anti-Iba1) for microglia activation was used to confirm probe location.

## Surgery and viral injections

Stereotaxic surgeries for viral transfection of ChR2 (AAV2-hsyn-ChR2-eYFP or AAV2-hsyn-ChR2-mCherry, University of North Carolina Vector Core Facility, virus titer 3.1 x 1012) were performed under isoflurane anesthesia (2%). Burr holes were drilled over the target location (GPe or striatum), and virus was injected using either a Nanoject (Drummond Scientific) and glass pulled pipette or a syringe pump (Harvard Scientific) fitted with a syringe (Hamilton) connected to PE10 tubing and a 30 gauge cannula. Viral injections were performed at p35-p50 and allowed to incubate for 2–4 weeks for optogenetic slice electrophysiology.

## Oscillation detection

Oscillating units units were detected by a two-step process as described in *Whalen et al., 2020*. First, we identified peaks in the 0.5– 4 Hz range of the power spectrum (computed with Welch's method and corrected for the unit's ISI distribution) and determined if any fell above a confidence interval estimated from high frequency (100– 500 Hz) power, correcting for multiple comparisons (Bonferroni correction). Then, to distinguish oscillations from 1/f noise, we determined if the mean phase shift at this identified frequency fell below a confidence interval estimated from high frequency phase shift. A unit which passed both these criteria was considered to be oscillating.

## Acknowledgements

This study was partially supported by NIH awards R01NS101016, R01NS104835, and R21NS095103 (AG) and NSF awards DMS 1516288 (AG, JR), 1612913 (JR), and 1724240 (JR). Some of the data incorporated into *Figure 12* was recorded in the Gittis lab by Kevin Mastro. We thank Tim Whalen for help processing the data for *Figure 12*, for discussions, and for comments on the manuscript.

## Additional information

### Competing interests

Aryn H Gittis: Reviewing editor, *eLife*. The other authors declare that no competing interests exist.

### Funding

| Funder | Grant reference number | Author |
|---|---|---|
| National Institutes of Health | R01NS101016 | Aryn H Gittis |
| National Science Foundation | 1516288 | Aryn H Gittis<br>Jonathan E Rubin |
| National Institutes of Health | R01NS104835 | Aryn H Gittis |
| National Institutes of Health | R21NS095103 | Aryn H Gittis |
| National Science Foundation | 1612913 | Jonathan E Rubin |
| National Science Foundation | 1724240 | Jonathan E Rubin |

The funders had no role in study design, data collection and interpretation, or the decision to submit the work for publication.

### Author contributions

Ryan S Phillips, Data curation, Software, Formal analysis, Investigation, Methodology, Writing - original draft, Writing - review and editing; Ian Rosner, Data curation, Investigation; Aryn H Gittis, Conceptualization, Resources, Data curation, Supervision, Funding acquisition, Investigation, Methodology, Writing - review and editing; Jonathan E Rubin, Conceptualization, Supervision, Funding acquisition, Methodology, Writing - original draft, Project administration, Writing - review and editing

### Author ORCIDs

Ryan S Phillips ⓘ https://orcid.org/0000-0002-8570-2348
Aryn H Gittis ⓘ http://orcid.org/0000-0002-3591-5775
Jonathan E Rubin ⓘ https://orcid.org/0000-0002-1513-1551

### Ethics

Animal experimentation: Experiments were conducted in accordance with the guidelines from the National Institutes of Health and with approval from Carnegie Mellon University Institutional Animal Care and Use Committee (protocol # AS15-018).

Decision letter and Author response
Decision letter https://doi.org/10.7554/eLife.55592.sa1
Author response https://doi.org/10.7554/eLife.55592.sa2

## Additional files

### Supplementary files

• Transparent reporting form

### Data availability

Data has been deposited on Dryad under https://doi.org/10.5061/dryad.tb2rbnzwx.

The following dataset was generated:

| Author(s) | Year | Dataset title | Dataset URL | Database and Identifier |
|---|---|---|---|---|
| Phillips R, Rosner I, Gittis AH, Rubin JE | 2020 | Mouse substantia nigra responses to optogenetic stimulation of projections from striatum and globus pallidus | https://doi.org/10.5061/dryad.tb2rbnzwx | Dryad Digital Repository, 10.5061/dryad.tb2rbnzwx |

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
