## [Decision Letter]

**Acceptance summary:**

This manuscript investigates the effect of GABAergic input on control of SNr activity, with a focus on how the shift in chloride reversal may change an inhibitory response to excitatory. It effectively combines experiments and modeling and spans cellular and network effects.

**Decision letter after peer review:**

Thank you for sending your article entitled "A computational model explains and predicts substantia nigra pars reticulata responses to pallidal and striatal inputs" for peer review at *eLife*. Your article is being evaluated by three peer reviewers, and the evaluation is being overseen by a Reviewing Editor and Kate Wassum as the Senior Editor.

Concerns about the extent of direct experimental evidence for the paper's conclusions were raised by both reviewers 1 and 2. Meanwhile, questions about the physiological relevance of the model's assumptions were raised by both reviewers 2 and 3. These criticisms are both central to the manuscript's claims. The possibility was raised that despite the lack of direct experimental evidence, the paper could be considered as primarily a modeling paper with some supporting experiments. However, it was agreed that for the modeling evidence to stand on its own as prediction and verification of likely changes in *E_Cl_* in biological neurons, a substantially more convincing bridge to biophysically relevant parameters needs to be demonstrated. Therefore, during the consultation process, there was consensus that there were enough uncertainties about both the extent of direct experimental evidence and of the physiological relevance of the model's assumptions that it was premature to move forward with publication at this stage. However, there is also quite a lot of positivity in the reviews, and the suggestions for new experiments and revisions are straightforward. Therefore, there appear to be several possible paths that successful revision might take.

In addition to the full reviews included below, the following points and suggestions were raised during consultation:

Regarding direct experimental evidence:

Gramicidin perforated patch-clamp experiments are hard but doable, and they could be used to show that E_GABA_ is changed following stimulation via both GPe and Str input. One might still see shifts in E_GABA_ in whole-cell mode, especially following dendritic input, as the a whole-cell patch can't effectively clamp Cl^–^ especially peripherally. An easier, more achievable experiment may be for them to repeat the cell attached experiments in the presence of selective KCC2 blockade (VU0463271 is the best), they should see a different distribution with more excitatory / biphasic responses etc. (There is a bit of controversy about pharmacological KCC2 enhancers – it's not clear those work).

Regarding Figure 6:

For the slice data shown in Figure 6 please report the number of mice, slices and neurons.

Were excitatory and/or cholinergic inputs pharmacologically blocked? Even if an inhibitory pathway was directly stimulated, in the time course shown, network effects could lead to excitation. Admittedly, the network effect candidates in slices are more limited than in vivo, but this should be ruled out.

Reviewer #1:

In their study "A computational model explains and predicts substantia nigra pars reticulata responses to pallidal and striatal inputs" Phillips et al. use computational models of substantia nigra pars reticulata (SNr) neurons which account for Cl^-^ dynamics to explain experimentally observed diversity in output responses following different GABAergic input. They then explore how Cl^-^ dynamics may account for how inhibitory input tunes the response properties of theses neurons under various conditions. They present in vivo data which is consistent with predictions from their model. I agree that the demonstration of biphasic responses are a good indication that Cl^-^ accumulation is occurring. In general, I am enthusiastic about this work, which uses computational modelling of Cl^-^ dynamics (which is often forgotten) to good effect to explain the diversity of experimental observations. I have no major comments for the authors to address.

Reviewer #2:

In this manuscript Phillips et al. examine the implications of depolarization in the chloride reversal potential (*E_Cl_*) in SNr neurons that could be triggered by chloride inflow due to tonic inhibitory input. Certainly SNr neurons are receiving a constant barrage of inhibitory inputs in vivo from GPe as well as striatum as described by the authors, and are a good candidate to ask these questions. The authors pursue a dual modeling and experimental approach to first argue for the likelihood that such depolarizing changes do occur, and second discover implications of such shifts on firing rates, slow oscillations, SNr synchronization, and changes in behaviourally relevant inhibitory responses. These finding are quite intriguing and the reviewer agrees with the authors that the results described are consistent with *E_Cl_* shifts performing important functional roles. However, throughout the entire set of simulations and experiments no direct evidence is brought forward towards the main hypothesis, and at each step alternative interpretations do exist. It is the opinion of this reviewer that such direct evidence needs to be delivered in order to make the study compelling, and that several types of experiments would be feasible to do so.

Major comments:

I will break these down into comments about experiments (A) and simulations (B).

A1) A direct demonstration of a shift of *E_Cl_* with GPe and Str input stimulation in slices should be made. A number of experiments could fully or partly deliver such evidence. As the authors indicate in the Discussion, perforated patching of SNr neurons might be ideal – and while not feasible in dendrites, it would be feasible on cell bodies. A bit simpler technically, and a little less powerful, would be to break into whole cell mode from the cell attached recording, and repeat stimulation after the chloride from the pipette controls *E_Cl_*. Excitatory stimulation effects and biphasic effects should disappear. In addition there are KCC2 blockers and enhancers available (see e.g. Hamidi and Avoli, 2015) and their use could shed light on the observed effects also. For instance, adding a KCC2 enhancer should shift biphasic responses towards pure inhibition.

A2) There are no experimental methods given at all for the in vivo data shown in Figure 10. It is not even clear if the mice were anesthetized or awake. A lot of the details seem to be taken from Whalen et al., 2020, but this paper is not published yet. A copy of the Whalen manuscript should be attached to the *eLife* submission. Is Figure 10 a part of this study? The disappearance of slow oscillations with GPe stimulation is clear, but quite a number of alternative explanations not related to *E_Cl_* in SNr exist for such a finding. To match the model more directly, it would be better to express halorhodopsin in the SNr, and directly control local chloride flow into the cell, instead of backfiring a large population of GPe axons that likely leads to network effects in the GPe and potentially STN.

B1) There is no statement in the manuscript that modeling scripts will be made available. For models to be replicable they need to be available and sharing is best done by posting on Yale ModelDB. This should be indicated in the manuscript.

B2) The model is a highly simplified 2-compartment model of SNr neurons. A careful match with the physiological properties of biological SNr neurons is not shown. Such a match is claimed in the text without evidence – this would be great material for a supplemental figure. (For instance, the spike cycle diagram in Figure 1D shows some obvious differences with the experimental diagram shown by Atherton et al., 2005. The AHP in Atherton et al. is -70 mV, and only -60 mV in the model, and the max *dv/dt* is over 200 in the data, and less than 150 in the model). The reviewer largely agrees that as a demonstration model of how *E_Cl_* shifts affect the responses to inhibition such detail may not be needed to match the data accurately. However, when it comes to the predictive power of the model with respect to time courses and levels of Cl concentration changes in the intracellular volume, a more detailed justification of how the volumes were chosen, and how the levels of KCC2 and Cl conductance are likely to map onto reality, should be given. The representation of multiple thin dendrites with a single lumped dendrite may impact such dynamics. Please discuss the appropriateness of the lumped dendrite for radial and axial chloride flow. With respect to Figure 11, how was the somato-dendritic coupling in terms of chloride flow chosen, and why would it match biophysical properties of SNr neurons?

B3) The use of 2 connected equal SNr model neurons to predict oscillations or synchrony in vivo seems poorly justified. Given the input from about 4 SNr neurons onto any given SNr neuron, a network of such sparsely connected neurons with varying delays in the axonal connection, as well as different basal firing rates plus some added noise might perform quite differently from the pair of connected neurons. To make more realistic predictions, it would be nice to see a network model of such heterogenous model neurons with realistic noise added as well.

Reviewer #3:

This manuscript investigates the effect of GABAergic input on control of SNr activity, with a focus on how the shift in chloride reversal may change an inhibitory response to excitatory. It is a great example of discoveries through synergistic interactions between modelers and experimentalists. The research spans cellular and network effects. The authors provide a fantastic explanation of the very difficult concept of PRC for single neurons. I especially appreciate Figure 10 – the in vivo test of effect of GPe stimulation, which has no effect on firing rate but suppresses oscillations. Overall it is a very well written manuscript and makes a significant contribution to our understanding of basal ganglia information processing.

Major concerns:

Figure 5: The authors need to comment on the relevance of change in effect, e.g. partial inhibition, over 1 sec given that SPNs do not fire at 20 Hz for 1 sec. Though a group of SPNs may indeed do that, these inputs would be distributed over multiple dendritic branches and therefore may not produce change in *E_Cl_*.

The PRC for coupled neurons is difficult to understand. In Figure 8, additional figure panels showing the traces and histogram for one or two cases would be helpful. Also, how similar do the two neurons need to be? What if coupled neurons are firing at different rates? This is especially important to show results when these neurons are firing closer to in vivo rates.

Figure 11: A single neuron mean firing rate > 20 Hz is not observed in the striatum. Perhaps the term "mean firing" refers to entire striatum and not to single neurons? If so, the authors need to use a different word. If not, it seems the model will not exhibit suppression for physiological rates. Again, this needs to be mentioned and put in context.

Regarding the coupling between soma and dendrite: given the length of dendrites, is 200ms a reasonable value for diffusion of chloride into the dendrite?

A repeated t-test is not the appropriate test here: "The spiking in each bin was then compared to baseline using t-tests where a p-value less than 0.05 was considered statistically significant." At the very least, correction for multiple t-tests is required. Ideally, a repeated measures ANOVA should be done.

---

## [Author Response]

In addition to the full reviews included below, the following points and suggestions were raised during consultation:Regarding direct experimental evidence:Gramicidin perforated patch-clamp experiments are hard but doable, and they could be used to show that E_GABA_ is changed following stimulation via both GPe and Str input. One might still see shifts in E_GABA_ in whole-cell mode, especially following dendritic input, as the a whole-cell patch can't effectively clamp Cl^-^ especially peripherally. An easier, more achievable experiment may be for them to repeat the cell attached experiments in the presence of selective KCC2 blockade (VU0463271 is the best), they should see a different distribution with more excitatory / biphasic responses etc. (There is a bit of controversy about pharmacological KCC2 enhancers – it's not clear those work).

As previously discussed with the editor, we could not perform additional experiments and focused our revision on the simulation and text editing steps in our work plan. In the final paragraph of our revised Discussion, we now mention several possible experiments that could be used to test ideas in this paper in future work, and we included the application of KCC2 blockers in this list.

Regarding Figure 6:For the slice data shown in Figure 6 please report the number of mice, slices and neurons.Were excitatory and/or cholinergic inputs pharmacologically blocked? Even if an inhibitory pathway was directly stimulated, in the time course shown, network effects could lead to excitation. Admittedly, the network effect candidates in slices are more limited than in vivo, but this should be ruled out.

The requested numbers of mice, slices and neurons have been added to the caption of Figure 6. Excitatory and cholinergic inputs were not blocked in these slice experiments. As the editor’s comment suggests, it is unlikely that such inputs contribute significantly to the activity observed in the slices used in these investigations. We have, however, added a sentence to the Discussion to point out that any such inputs that were present could have contributed to SNr responses:

“A final consideration relating to our slice experiments is that we did not block excitatory or cholinergic inputs. Thus, related network effects theoretically could have contributed to the SNr responses, although there are no known sources for such effects in the slices that we studied.”

Reviewer #2:In this manuscript Phillips et al. examine the implications of depolarization in the chloride reversal potential (E_Cl_) in SNr neurons that could be triggered by chloride inflow due to tonic inhibitory input. Certainly SNr neurons are receiving a constant barrage of inhibitory inputs in vivo from GPe as well as striatum as described by the authors, and are a good candidate to ask these questions. The authors pursue a dual modeling and experimental approach to first argue for the likelihood that such depolarizing changes do occur, and second discover implications of such shifts on firing rates, slow oscillations, SNr synchronization, and changes in behaviorally relevant inhibitory responses. These finding are quite intriguing and the reviewer agrees with the authors that the results described are consistent with E_Cl_ shifts performing important functional roles. However, throughout the entire set of simulations and experiments no direct evidence is brought forward towards the main hypothesis, and at each step alternative interpretations do exist. It is the opinion of this reviewer that such direct evidence needs to be delivered in order to make the study compelling, and that several types of experiments would be feasible to do so.

Unfortunately, the shutdown of laboratories due to COVID-19 has precluded the performance of additional experiments. We have added some material to the Discussion of the manuscript to mention ideas for future experimental tests, inspired by the comments of all of the reviewers. We have tested model robustness to a wide variety of features through extensive new simulations, which we describe in more detail below.

Major comments:I will break these down into comments about experiments (A) and simulations (B).A1) A direct demonstration of a shift of E_Cl_ with GPe and Str input stimulation in slices should be made. A number of experiments could fully or partly deliver such evidence. As the authors indicate in the Discussion, perforated patching of SNr neurons might be ideal – and while not feasible in dendrites, it would be feasible on cell bodies. A bit simpler technically, and a little less powerful, would be to break into whole cell mode from the cell attached recording, and repeat stimulation after the chloride from the pipette controls E_Cl_. Excitatory stimulation effects and biphasic effects should disappear. In addition there are KCC2 blockers and enhancers available (see e.g. Hamidi and Avoli, 2015) and their use could shed light on the observed effects also. For instance, adding a KCC2 enhancer should shift biphasic responses towards pure inhibition.

The reviewer presents some excellent suggestions for future experiments, some of which were indeed on our to-do lists when we lost lab access. We have mentioned these in the following text added to the final Discussion paragraph:

“A more attainable first step would be to repeat the in vitro stimulation experiments under pharmacological blockade of KCC2, to check for a resulting bias toward excitatory and biphasic responses, and in the presence of KCC2 enhancers, to check for a shift toward inhibitory responses (Hamidi and Avoli, 2015). Another option would be to break into whole cell mode and repeat stimulation with control of *E_Cl_*using the chloride from the pipette, to check if excitatory and biphasic effects can be eliminated… ”

A2) There are no experimental methods given at all for the in vivo data shown in Figure 10. It is not even clear if the mice were anesthetized or awake. A lot of the details seem to be taken from Whalen et al., 2020, but this paper is not published yet. A copy of the Whalen manuscript should be attached to the eLife submission. Is Figure 10 a part of this study? The disappearance of slow oscillations with GPe stimulation is clear, but quite a number of alternative explanations not related to E_Cl_ in SNr exist for such a finding. To match the model more directly, it would be better to express halorhodopsin in the SNr, and directly control local chloride flow into the cell, instead of backfiring a large population of GPe axons that likely leads to network effects in the GPe and potentially STN.

We apologize for the oversight in not providing the methods for the in vivo data appearing in Figure 10. A new in vivo electrophysiology section has been added to Materials and methods. The Whalen paper has now been published and an updated citation has been provided. Finally, the suggestion of using halorhodopsin is an excellent one, and we have extended the Discussion text mentioned above to conclude with:

“…eliminated. A final option is to express halorhodopsin in the SNr, and directly control local chloride flow into the cell.”

B1) There is no statement in the manuscript that modeling scripts will be made available. For models to be replicable they need to be available and sharing is best done by posting on Yale ModelDB. This should be indicated in the manuscript.

We will certainly make our modeling scripts freely available.

B2) The model is a highly simplified 2-compartment model of SNr neurons. A careful match with the physiological properties of biological SNr neurons is not shown. Such a match is claimed in the text without evidence – this would be great material for a supplemental figure. (For instance, the spike cycle diagram in Figure 1D shows some obvious differences with the experimental diagram shown by Atherton et al., 2005. The AHP in Atherton et al. is -70 mV, and only -60 mV in the model, and the max dv/dt is over 200 in the data, and less than 150 in the model).

The AHP in the model was tuned to match traces shown in (Higgs and Wilson, 2016). The text at the end of the Results section on the SNr model now reads,

“The baseline firing rate (≈ 10*Hz*) and action potential peak of the model are tuned to match experimental data from in vitro mouse and rat slice recordings (Richards et al., 1997; Atherton and Bevan, 2005; Yanovsky et al., 2006; Zhou et al., 2008; Ding et al., 2011), while the AHP was tuned to match data presented in (Higgs and Wilson, 2016). For a full model description see Materials and methods.”

We acknowledge that the maximum of *dv/dt* in our model is less than that in the data of Atherton et al. However, while the difference between 150*mV/ms* and 200*mV/ms* is large, these extreme values are achieved for an extremely brief time. Thus, this difference results in a spike width difference between the model and the real neuron of only a small fraction of a millisecond.

The reviewer largely agrees that as a demonstration model of how E_Cl_ shifts affect the responses to inhibition such detail may not be needed to match the data accurately. However, when it comes to the predictive power of the model with respect to time courses and levels of Cl concentration changes in the intracellular volume, a more detailed justification of how the volumes were chosen, and how the levels of KCC2 and Cl conductance are likely to map onto reality, should be given. The representation of multiple thin dendrites with a single lumped dendrite may impact such dynamics. Please discuss the appropriateness of the lumped dendrite for radial and axial chloride flow. With respect to Figure 11, how was the somato-dendritic coupling in terms of chloride flow chosen, and why would it match biophysical properties of SNr neurons?

We have thoroughly addressed these points.

B3) The use of 2 connected equal SNr model neurons to predict oscillations or synchrony in vivo seems poorly justified. Given the input from about 4 SNr neurons onto any given SNr neuron, a network of such sparsely connected neurons with varying delays in the axonal connection, as well as different basal firing rates plus some added noise might perform quite differently from the pair of connected neurons. To make more realistic predictions, it would be nice to see a network model of such heterogenous model neurons with realistic noise added as well.

We have explored the effects of synaptic delays, in vivo SNr firing rates, the inclusion of noise, differing firing rates between pre- and postsynaptic SNr neurons, multiple presynaptic neurons as well as oscillations in a sparsely connected network of SNr neurons.

Reviewer #3:[…]Major concerns:Figure 5: The authors need to comment on the relevance of change in effect, e.g. partial inhibition, over 1 sec given that SPNs do not fire at 20 Hz for 1 sec. Though a group of SPNs may indeed do that, these inputs would be distributed over multiple dendritic branches and therefore may not produce change in E_Cl_.

The data in Figure 5 are meant to predict the effects of optogenetic stimulation of Str terminals in the SNr, which forces synchronous synaptic activation of a large portion of Str terminals at 20*Hz*. As the reviewer points out, individual SPNs cannot sustain this firing rate for 1*s* and therefore, this experiment may not represent a biologically relevant scenario. However, the same net Str input could be elicited by multiple Str neurons firing at a slower frequency and would generate the same SNr firing rate response and *Cl*^−^ dynamics. It is also possible that these inputs are spread out over many dendritic branches. In this case the *Cl*^−^ may vary across branches with some that do not change at all and others that become very depolarized; however, the average *Cl*^−^ change would likely still resemble those shown in Figure 5. In revision, we have added the following sentences on this point to the Discussion:

“We note that these Str neuron baseline firing rates are significantly lower than the stimulation frequency in our optogenetic activation of Str terminals. Nonetheless, similar *Cl*^−^ dynamics and SNr responses could result from a collection of lower-rate Str inputs in natural settings, albeit with heterogeneity over specific levels of *Cl*^−^ across different dendritic branches.”

The PRC for coupled neurons is difficult to understand. In Figure 8, additional figure panels showing the traces and histogram for one or two cases would be helpful. Also, how similar do the two neurons need to be? What if coupled neurons are firing at different rates? This is especially important to show results when these neurons are firing closer to in vivo rates.

To clarify Figure 8 and to complement the information already shown in Figure 8 as well as in the particular case in Figure 9, we have made some edits to Figure 8. Specifically, we have included histograms in each of panels A1-A4 and B1-B4 to illustrate the phases of neuron 2 when its inputs from neuron 1 arrive. We have also changed the axis labels to clarify the interpretation of these plots. Finally, we have added a new figure, Figure 8—figure supplement 1, to illustrate how the PRC works, and promotes anti-phase locking, for bidirectionally coupled neurons.

We have performed new simulations to explore how our results on synchrony and oscillations depend on heterogeneity in SNr firing rates and on the baseline SNr firing rate (specifically, how results persist under increases to in vivo rates). These results are reported the revised manuscript and supplemental materials.

Figure 11: A single neuron mean firing rate > 20 Hz is not observed in the striatum. Perhaps the term "mean firing" refers to entire striatum and not to single neurons? If so, the authors need to use a different word. If not, it seems the model will not exhibit suppression for physiological rates. Again, this needs to be mentioned and put in context.

The reviewer is correct that the striatal firing rate indicated in the old Figure 11D (now Figure 13D) refers to the total rate of striatal input to the postsynaptic SNr cell. We have changed the y-axis label of that figure panel accordingly.

Regarding the coupling between soma and dendrite: given the length of dendrites, is 200ms a reasonable value for diffusion of chloride into the dendrite?

This comment relates to the following text in the Materials and methods section of our manuscript: “Additionally, diffusion of *Cl*^−^ between the somatic and dendritic compartments is incorporated into the network model. This was simulated by the addition of the exponential decay terms – ([Cl]Sin−[Cl]Din)/(TSD) and –([Cl]Sin−[Cl]Sin)/(TDS) into [the differential equation for intracellular chloride] for the somatic and dendritic compartments respectively. The parameters *τ_SD_*= 200*ms* and *τ_DS_*= 80*ms* are exponential decay time constants.”

Our data suggests that *E_GABA_*is more depolarized in the soma than in the dendrite. This difference is likely due to the higher chloride load in the soma from tonic GPe inputs as well as the preferential expression of KCC2 in dendrites of SNr neurons (Gulacsi et al., 2003). Because of this configuration, it is likely that somatic *Cl*^−^ will diffuse from the soma to the dendrite, ultimately affecting dendritic *E_GABA_*. This is the motivation for the somato-dendritic *Cl* coupling term, which is relevant for the simulations shown in Figure 11 (now Figure 13) of the main manuscript. The specific time constants, *τ_SD_*and *τ_DS_*, were set at values for which *E_GABA_*in the dendrite was hyperpolarized relative to the soma by approximately 2.5 − 8*mV* , as reported in the literature (Connelly et al., 2010; Lavian and Korngreen, 2016) (see Figure 13C in the main manuscript). The main point of somato-dendritic *Cl* coupling in Figure 13 is to illustrate the idea that changes in the somatic *Cl* load (presumably due to changes in GPe firing rates) could shift dendritic *E_GABA_*and, therefore, the strength of Str inhibition. The specific values of *τ_SD_*and *τ_DS_*are not intended to reflect or match rates of axial *Cl* diffusion.

We have added part of this explanation to the Materials and methods section in the main manuscript to clarify these points for readers.

A repeated t-test is not the appropriate test here: "The spiking in each bin was then compared to baseline using t-tests where a p-value less than 0.05 was considered statistically significant." At the very least, correction for multiple t-tests is required. Ideally, a repeated measures ANOVA should be done.

Thanks for pointing out this issue. Since we were not confident that our data satisfies the assumptions needed for ANOVA, we repeated this statistical analysis with a Mann–Whitney U test with Bonferroni correction for multiple comparisons, where *p <* 0.00416 was considered statistically significant. We replaced Figure 6 and its supplemental figures (Figure 6—figure supplements 1 and 2) with new versions based on the corrected analysis, which did not change any of our conclusions.